# Boosting Barely Robust Learners:
# A New Perspective on Adversarial Robustness

**Avrim Blum**
avrim@ttic.edu
TTI Chicago

**Omar Montasser**
omar@ttic.edu
TTI Chicago

**Greg Shakhnarovich**
greg@ttic.edu
TTI Chicago

**Hongyang Zhang**
first.last@uwaterloo.ca
University of Waterloo

## Abstract

We present an oracle-efficient algorithm for boosting the *adversarial robustness* of *barely robust* learners. Barely robust learning algorithms learn predictors that are adversarially robust only on a small fraction $\beta \ll 1$ of the data distribution. Our proposed notion of barely robust learning requires robustness with respect to a "larger" perturbation set; which we show is *necessary* for strongly robust learning, and that weaker relaxations are *not* sufficient for strongly robust learning. Our results reveal a qualitative and quantitative *equivalence* between two seemingly unrelated problems: strongly robust learning and barely robust learning.

## 1 Introduction

We consider the problem of learning predictors that are *robust* to adversarial examples at test time. That is, we would like to be robust against a perturbation set $\mathcal{U} : \mathcal{X} \to 2^{\mathcal{X}}$, where $\mathcal{U}(x) \subseteq \mathcal{X}$ is the set of allowed perturbations that an adversary might replace $x$ with, e.g., $\mathcal{U}$ could be perturbations of bounded $\ell_p$-norms [10]. The goal is to learn a predictor $h$ with small *robust risk*:

$$\mathrm{R}_{\mathcal{U}}(h; \mathcal{D}) \triangleq \Pr_{(x,y) \sim \mathcal{D}} \left[ \exists z \in \mathcal{U}(x) : h(z) \neq y \right]. \tag{1}$$

Adversarially robust learning has proven to be quite challenging in practice, where current adversarial learning methods typically learn predictors with low natural error but robust only on a small fraction of the data. For example, according to the RobustBench leaderboard [7], the highest achieved robust accuracy with respect to $\ell_\infty$ perturbations on CIFAR10 is $\approx 66\%$ and on ImageNet is $\approx 38\%$. Can we leverage existing methods and go beyond their limits? This motivates us to pursue the idea of *boosting* robustness, and study the following theoretical question:

*Can we boost* barely *robust learning algorithms to learn predictors with high* robust *accuracy?*

That is, given a *barely* robust learning algorithm $\mathbb{A}$ which can only learn predictors robust on say $\beta = 10\%$ fraction of the data distribution, we are asking whether it is possible to *boost* the robustness of $\mathbb{A}$ and learn predictors with high *robust* accuracy, say $90\%$. We want to emphasize that we are interested here in extreme situations when the robustness parameter $\beta \ll 1$. We are interested in generic boosting algorithms that take as input a black-box learner $\mathbb{A}$ and a specification of the perturbation set $\mathcal{U}$, and output a predictor with high robust accuracy by repeatedly calling $\mathbb{A}$.

In this work, by studying the question above, we offer a new perspective on adversarial robustness. Specifically, we discover a qualitative and quantitative *equivalence* between two seemingly unrelated problems: strongly robust learning and barely robust learning. We show that barely robust learning implies strongly robust learning through a novel algorithm for *boosting* robustness. As we elaborate below, our proposed notion of barely robust learning requires robustness with respect to a "larger" perturbation set. We also show that this is *necessary* for strongly robust learning, and that weaker relaxations of barely robust learning do not imply strongly robust learning.

36th Conference on Neural Information Processing Systems (NeurIPS 2022).

## 1.1 Main contributions

When formally studying the problem of boosting robustness, an important question emerges which is: what notion of "barely robust" learning is required for boosting robustness? As we shall show, this is not immediately obvious. One of the main contributions of this work is the following key definition of *barely robust* learners:

**Definition 1** (Barely Robust Learner). Learner $\mathbb{A}$ $(\beta, \epsilon, \delta)$-barely-robustly-learns a concept $c : \mathcal{X} \to \mathcal{Y}$ w.r.t. $\mathcal{U}^{-1}(\mathcal{U})$ if $\exists m_{\mathbb{A}}(\beta, \epsilon, \delta) \in \mathbb{N}$ s.t. for any distribution $D$ over $\mathcal{X}$ s.t. $\Pr_{x \sim D} [\exists z \in \mathcal{U}(x) : c(z) \neq c(x)] = 0$, with prob. at least $1 - \delta$ over $S = \{(x_i, c(x_i))\}_{i=1}^{m} \sim D_c$ and any internal randomness in the learner, $\mathbb{A}$ outputs a predictor $\hat{h} = \mathbb{A}(S)$ satisfying:

$$\Pr_{x \sim D} \left[ \forall \tilde{x} \in \mathcal{U}^{-1}(\mathcal{U})(x) : \hat{h}(\tilde{x}) = \hat{h}(x) \right] \geq \beta \quad \text{and} \quad \Pr_{x \sim D} \left[ \hat{h}(x) \neq c(x) \right] \leq \epsilon.$$

Notice that we require $\beta$-robustness with respect to a "larger" perturbation set $\mathcal{U}^{-1}(\mathcal{U})$. Specifically, $\mathcal{U}^{-1}(\mathcal{U})(x)$ is the set of all *natural* examples $\tilde{x}$ that share an adversarial perturbation $z$ with $x$ (see Equation 2). E.g., if $\mathcal{U}(x)$ is an $\ell_p$-ball with radius $\gamma$, then $\mathcal{U}^{-1}(\mathcal{U})(x)$ is an $\ell_p$-ball with radius $2\gamma$.

On the other hand, $(\epsilon, \delta)$-robustly-learning a concept $c$ with respect to $\mathcal{U}$ is concerned with learning a predictor $\hat{h}$ from samples $S$ with small robust risk $\mathrm{R}_{\mathcal{U}}(\hat{h}; D_c) \leq \epsilon$ with probability at least $1 - \delta$ over $S \sim D_c^m$ (see Equation 1 and Definition 2), where we are interested in robustness with respect to $\mathcal{U}$ and *not* $\mathcal{U}^{-1}(\mathcal{U})$. Despite this qualitative difference between *barely robust* learning and *strongly robust* learning, we provably show next that they are in fact *equivalent*.

Our main algorithmic result is $\beta$-RoBoost, an oracle-efficient boosting algorithm that boosts barely robust learners to strongly robust learners:

**Theorem 1.** *For any perturbation set $\mathcal{U}$, $\beta$-RoBoost $(\epsilon, \delta)$-robustly-learns any target concept $c : \mathcal{X} \to \mathcal{Y}$ w.r.t. $\mathcal{U}$ using $T = \frac{\ln(2/\epsilon)}{\beta}$ black-box oracle calls to any $(\beta, \frac{\beta\epsilon}{2}, \frac{\delta}{2T})$-barely-robust learner $\mathbb{A}$ for $c$ w.r.t. $\mathcal{U}^{-1}(\mathcal{U})$, with sample complexity $\frac{4Tm_{\mathbb{A}}}{\epsilon}$, where $m_{\mathbb{A}}$ is the sample complexity of learner $\mathbb{A}$.*

The result above shows that barely robust learning is *sufficient* for strongly robust learning. An important question remains, however: is our proposed notion of barely robust learning *necessary* for strongly robust learning? In particular, our proposed notion of barely robust learning requires $\beta$-robustness with respect to a "larger" perturbation set $\mathcal{U}^{-1}(\mathcal{U})$, instead of the actual perturbation set $\mathcal{U}$ that we care about. We provably show next that this is *necessary*.

**Theorem 4.** *For any $\mathcal{U}$, learner $\mathbb{B}$, and $\epsilon \in (0, 1/4)$, if $\mathbb{B}$ $(\epsilon, \delta)$-robustly-learns some unknown target concept $c$ w.r.t. $\mathcal{U}$, then there is a learner $\tilde{\mathbb{B}}$ that $(\frac{1-\epsilon}{2}, 2\epsilon, 2\delta)$-barely-robustly-learns $c$ w.r.t. $\mathcal{U}^{-1}(\mathcal{U})$.*

This still does *not* rule out the possibility that boosting robustness is possible even with the weaker requirement of $\beta$-robustness with respect to $\mathcal{U}$. But we show next that, indeed, barely robust learning with respect to $\mathcal{U}$ is *not sufficient* for strongly robust learning with respect to $\mathcal{U}$:

**Theorem 6.** *There is a space $\mathcal{X}$, a perturbation set $\mathcal{U}$, and a class of concepts $\mathcal{C}$ s.t. $\mathcal{C}$ is $(\beta = \frac{1}{2}, \epsilon = 0, \delta)$-barely-robustly-learnable w.r.t $\mathcal{U}$, but $\mathcal{C}$ is not $(\epsilon, \delta)$-robustly-learnable w.r.t. $\mathcal{U}$ for any $\epsilon < 1/2$.*

Our results offer a new perspective on adversarially robust learning. We show that two seemingly unrelated problems: barely robust learning w.r.t. $\mathcal{U}^{-1}(\mathcal{U})$ and strongly robust learning w.r.t. $\mathcal{U}$, are in fact *equivalent*. The following corollary follows from Theorem 1 and Theorem 4.

**Corollary I.** *For any class $\mathcal{C}$ and any perturbation set $\mathcal{U}$, $\mathcal{C}$ is strongly robustly learnable with respect to $\mathcal{U}$ if and only if $\mathcal{C}$ is barely robustly learnable with respect to $\mathcal{U}^{-1}(\mathcal{U})$.*

We would like to note that having a separate robustness parameter $\beta$ and a natural error parameter $\epsilon$ allows us to consider regimes where $\beta < \frac{1}{2}$ and $\epsilon$ is small. This models typical scenarios in practice where learning algorithms are able to learn predictors with reasonably low natural error but the predictors are only barely robust. More generally, this allows us to explore the relationship between $\beta$ and $\epsilon$ in terms of boosting robustness (see Section 5 for a more elaborate discussion).

**Landscape of boosting robustness.** Our results reveal an interesting landscape for boosting robustness when put in context of prior work. When the robustness parameter $\beta > \frac{1}{2}$, it is known from prior work that $\beta$-robustness with respect to $\mathcal{U}$ *suffices* for boosting robustness [see e.g., 15, 1], which

is witnessed by the $\alpha$-Boost algorithm [18]. When the robustness parameter $\beta \leq \frac{1}{2}$, our results show that boosting is still *possible*, but $\beta$-robustness with respect to $\mathcal{U}^{-1}(\mathcal{U})$ is *necessary* and we *cannot* boost robustness with $\beta$-robustness with respect to $\mathcal{U}$.

In fact, by combining our algorithm $\beta$-RoBoost with $\alpha$-Boost, we obtain an even *stronger* boosting result that only requires barely robust learners with a natural error parameter that does *not* scale with the targeted robust error. Beyond that, our results imply that we can even boost robustness with respect to $\mathcal{U}^{-1}(\mathcal{U})$. This is summarized in the following corollary which follows from Theorem 1, Lemma 8, and Theorem 4.

**Corollary II** (Landscape of Boosting Robustness). *Let $\mathcal{C}$ be a class of concepts. For fixed $\epsilon_0, \delta_0 = \left(\frac{1}{3}, \frac{1}{3}\right)$ and any target $\epsilon < \epsilon_0$ and $\delta > \delta_0$:*

1. *If $\mathcal{C}$ is $(\beta, \frac{\beta\epsilon_0}{2}, \frac{\beta\delta_0}{\ln(2/\epsilon_0)})$-barely-robustly-learnable w.r.t. $\mathcal{U}^{-1}(\mathcal{U})$, then $\mathcal{C}$ is $(\epsilon_0, \delta_0)$-robustly-learnable w.r.t. $\mathcal{U}$.*

2. *If $\mathcal{C}$ is $(\epsilon_0, \delta_0)$-robustly-learnable w.r.t. $\mathcal{U}$, then $\mathcal{C}$ is $(\epsilon, \delta)$-robustly learnable w.r.t. $\mathcal{U}$.*

3. *If $\mathcal{C}$ is $(\epsilon, \delta)$-robustly learnable w.r.t. $\mathcal{U}$, then $\mathcal{C}$ is $(\frac{1-\epsilon}{2}, 2\epsilon, 2\delta)$-barely-robustly-learnable w.r.t. $\mathcal{U}^{-1}(\mathcal{U})$.*

*In particular, $1 \Rightarrow 2 \Rightarrow 3$ reveals that we can also algorithmically boost robustness w.r.t. $\mathcal{U}^{-1}(\mathcal{U})$.*

**Boosting robustness with unlabeled data.**    In Appendix A and Theorem 7, we show that a variant of our boosting algorithm, $\beta$-URoBoost, can boost robustness using *unlabeled* data when having access to a barely robust learner $\mathbb{A}$ that is tolerant to small noise in the labels. In Appendix E, we discuss an idea of obtaining robustness at different levels of granularity through our boosting algorithm. Specifically, when $\mathcal{U}(x)$ is a metric-ball around $x$ with radius $\gamma$, we can learn a predictor $\hat{h}$ with different robustness levels: $\gamma, \frac{\gamma}{2}, \frac{\gamma}{4}, \ldots$, in different regions of the distribution.

## 1.2  Related work

To put our work in context, the classic and pioneering works of [12, 17, 8, 9] explored the question of boosting the *accuracy* of *weak* learning algorithms, from accuracy slightly better than $\frac{1}{2}$ to arbitrarily high accuracy in the realizable PAC learning setting. Later works have explored boosting the accuracy in the agnostic PAC setting [see e.g., 11].

In this work, we are interested in the problem of boosting *robustness* rather than accuracy. Specifically, boosting robustness of learners $\mathbb{A}$ that are highly accurate on *natural* examples drawn from the data distribution, but robust only on some $\beta$ fraction of them. We consider this problem in the *robust realizable* setting, that is, when the unknown target concept $c$ has zero robust risk $\mathrm{R}_\mathcal{U}(c; D_c) = 0$. For example, with linear classifiers and $\ell_2$ perturbations, robust realizability is equivalent to linear separability with a margin, which is a setting that is studied extensively in the literature. Furthermore, for vision datasets in practice (e.g., CIFAR10 and ImageNet), it has been observed that the classes can be robustly separated with some non-negligible distance, indicating that robust classifiers for these datasets do exist, i.e., the robust realizability assumption holds [see Section 3 in 19].

We know from prior work [see e.g., 15, 1, 20] that if the robustness parameter $\beta > \frac{1}{2}$, then barely robust learning w.r.t. $\mathcal{U}$ implies strongly robust learning w.r.t. $\mathcal{U}$. In this case, we essentially have *weak* learners w.r.t. the robust risk $\mathrm{R}_\mathcal{U}$, and the original boosting algorithms such as the $\alpha$-Boost algorithm [18, Section 6.4.2] can boost the robust risk. In this work, we focus on boosting *barely* robust learners, i.e., mainly when the robustness parameter $\beta \leq 1/2$, but our algorithm works for any $0 < \beta \leq 1$.

[15] studied the problem of adversarially robust learning (as in Definition 2). They showed that if a hypothesis class $\mathcal{C}$ is PAC learnable non-robustly (i.e., $\mathcal{C}$ has finite VC dimension), then $\mathcal{C}$ is adversarially robustly learnable. This result, however, is not constructive and the robust learning algorithm given does not directly use a black-box non-robust learner. Later on, [16] studied a more constructive version of the same question: reducing strongly robust learning to *non-robust* PAC learning when given access to black-box *non-robust* PAC learners. This is different from the question we study in this work. In particular, we explore the relationship between strongly robust learning and barely robust learning, and we present a boosting algorithm for learners that already have some non-trivial robustness guarantee $\beta > 0$.

## 2 Preliminaries

Let $\mathcal{X}$ denote the instance space and $\mathcal{Y}$ denote the label space. We would like to be robust with respect to a perturbation set $\mathcal{U} : \mathcal{X} \to 2^{\mathcal{X}}$, where $\mathcal{U}(x) \subseteq \mathcal{X}$ is the set of allowed adversarial perturbations that an adversary might replace $x$ with at test time. Denote by $\mathcal{U}^{-1}$ the inverse image of $\mathcal{U}$, where for each $z \in \mathcal{X}$, $\mathcal{U}^{-1}(z) = \{x \in \mathcal{X} : z \in \mathcal{U}(x)\}$. Observe that for any $x, z \in \mathcal{X}$ it holds that $z \in \mathcal{U}(x) \Leftrightarrow x \in \mathcal{U}^{-1}(z)$. Furthermore, when $\mathcal{U}$ is symmetric, where for any $x, z \in \mathcal{X}, z \in \mathcal{U}(x) \Leftrightarrow x \in \mathcal{U}(z)$, it holds that $\mathcal{U} = \mathcal{U}^{-1}$. For each $x \in \mathcal{X}$, denote by $\mathcal{U}^{-1}(\mathcal{U})(x)$ the set of all *natural* examples $\tilde{x}$ that share some adversarial perturbation $z$ with $x$, i.e.,

$$\mathcal{U}^{-1}(\mathcal{U})(x) = \cup_{z \in \mathcal{U}(x)} \mathcal{U}^{-1}(z) = \{\tilde{x} : \exists z \in \mathcal{U}(x) \cap \mathcal{U}(\tilde{x})\}. \tag{2}$$

For example, when $\mathcal{U}(x) = \mathrm{B}_\gamma(x) = \{z \in \mathcal{X} : \rho(x, z) \le \gamma\}$ where $\gamma > 0$ and $\rho$ is some metric on $\mathcal{X}$ (e.g., $\ell_p$-balls), then $\mathcal{U}^{-1}(\mathcal{U})(x) = \mathrm{B}_{2\gamma}(x)$. For any classifier $h : \mathcal{X} \to \mathcal{Y}$ and any $\mathcal{U}$, denote by $\mathrm{Rob}_{\mathcal{U}}(h)$ the *robust region* of $h$ with respect to $\mathcal{U}$ defined as:

$$\mathrm{Rob}_{\mathcal{U}}(h) \triangleq \{x \in \mathcal{X} : \forall z \in \mathcal{U}(x), h(z) = h(x)\}. \tag{3}$$

**Definition 2** (Strongly Robust Learner). *Learner $\mathbb{B}$ $(\epsilon, \delta)$-robustly-learns a concept $c : \mathcal{X} \to \mathcal{Y}$ with respect to $\mathcal{U}$ if there exists $m(\epsilon, \delta) \in \mathbb{N}$ s.t. for any distribution $D$ over $\mathcal{X}$ satisfying $\Pr_{x \sim D} [\exists z \in \mathcal{U}(x) : c(z) \ne c(x)] = 0$, with probability at least $1 - \delta$ over $S = \{(x_i, c(x_i))\}_{i=1}^m \sim D_c$, $\mathbb{B}$ outputs a predictor $\hat{h} = \mathbb{B}(S)$ satisfying:*

$$\mathrm{R}_{\mathcal{U}}(\hat{h}; D_c) = \Pr_{x \sim D} \left[\exists z \in \mathcal{U}(x) : \hat{h}(z) \ne c(x)\right] \le \epsilon.$$

## 3 Boosting a barely robust Learner to a strongly robust learner

We present our main result in this section: $\beta$-RoBoost is an algorithm for *boosting* the *robustness* of *barely* robust learners. Specifically, in Theorem 1, we show that given a *barely* robust learner $\mathbb{A}$ for some unknown target concept $c$ (according to Definition 1), it is *possible* to strongly robustly learn $c$ with $\beta$-RoBoost by making black-box oracle calls to $\mathbb{A}$.

---

**Algorithm 1:** $\beta$-RoBoost — Boosting *barely* robust learners.

**Input:** Sampling oracle for distribution $D_c$, black-box $(\beta, \epsilon, \delta)$-barely-robust learner $\mathbb{A}$.

1 Set $T = \frac{\ln(2/\epsilon)}{\beta}$, and $m = \max \left\{ m_{\mathbb{A}}(\beta, \frac{\beta\epsilon}{2}, \frac{\delta}{2T}), 4 \ln \left(\frac{2T}{\delta}\right) \right\}$.

2 **while** $1 \le t \le T$ **do**

3      Call RejectionSampling on $h_1, \ldots, h_{t-1}$ and $m$, and let $\tilde{S}_t$ be the returned dataset.

4      **If** $\tilde{S}_t \ne \emptyset$, **then** call learner $\mathbb{A}$ on $\tilde{S}_t$ and let $h_t = \mathbb{A}(\tilde{S}_t)$ be its output. **Otherwise**, break.

     **Output:** The cascade predictor defined as

         $\mathrm{CAS}(h_1, \ldots, h_T)(z) \triangleq G_{h_s}(z)$ where $s = \min \{1 \le t \le T : G_{h_t}(z) \ne \perp\}$, and

         *selective classifiers* $\quad G_{h_t}(z) \triangleq \begin{cases} y, & \text{if } (\exists y \in \mathcal{Y}) (\forall \tilde{x} \in \mathcal{U}^{-1}(z)) : h(\tilde{x}) = y; \\ \perp, & \text{otherwise.} \end{cases}$

5 RejectionSampling(*predictors $h_1, \ldots, h_t$, and sample size $m$*):

6      **for** $1 \le i \le m$ **do**

7          Draw samples $(x, y) \sim D$ until sampling an $(x_i, y_i)$ s.t.: $\forall_{t' \le t} \exists_{z \in \mathcal{U}(x_i)} G_{h_{t'}}(z) = \perp$.
             `// sampling from the region of D where all predictors `$h_1, \ldots, h_t$` are not robust.`

8          If this costs more than $\frac{4}{\epsilon}$ samples from $D$, abort and return an empty dataset $\tilde{S} = \emptyset$.
             `// If the mass of the non-robust region is small, then we can safely terminate.`

9      Output dataset $\tilde{S} = \{(x_1, y_1), \ldots, (x_m, y_m)\}$.

---

**Theorem 1.** *For any perturbation set $\mathcal{U}$, $\beta$-RoBoost $(\epsilon, \delta)$-robustly-learns any target concept $c$ w.r.t. $\mathcal{U}$ using $T = \frac{\ln(2/\epsilon)}{\beta}$ black-box oracle calls to any $(\beta, \frac{\beta\epsilon}{2}, \frac{\delta}{2T})$-barely-robust learner $\mathbb{A}$ for $c$ w.r.t. $\mathcal{U}^{-1}(\mathcal{U})$, with total sample complexity*

$$m(\epsilon, \delta) \le \frac{4T \max \left\{ m_{\mathbb{A}}(\beta, \frac{\beta\epsilon}{2}, \frac{\delta}{2T}), 4 \ln \left(\frac{2T}{\delta}\right) \right\}}{\epsilon}.$$

In fact, we present next an even stronger result for boosting $(\beta, \epsilon_0, \delta_0)$-barely-robust-learners with fixed error $\epsilon_0 = \frac{\beta}{6}$ and confidence $\delta_0 = \frac{\beta}{6\ln(6)}$. This is established by combining two boosting algorithms: $\beta$-RoBoost from Theorem 1 and $\alpha$-Boost from earlier work [see e.g., 15, 18] which is presented in Appendix B for convenience. The main idea is to perform two layers of boosting. In the first layer, we use $\beta$-RoBoost to get a $(\frac{1}{3}, \frac{1}{3})$-robust-learner w.r.t. $\mathcal{U}$ from a $(\beta, \epsilon_0, \delta_0)$-barely-robust-learner $\mathbb{A}$ w.r.t. $\mathcal{U}^{-1}(\mathcal{U})$. Then, in the second layer, we use $\alpha$-Boost to boost $\beta$-RoBoost from a $(\frac{1}{3}, \frac{1}{3})$-robust-learner to an $(\epsilon, \delta)$-robust-learner w.r.t. $\mathcal{U}$.

**Corollary 2** (Stronger Boosting Guarantee). *For any perturbation set $\mathcal{U}$, $\alpha$-Boost combined with $\beta$-RoBoost $(\epsilon, \delta)$-robustly-learn any target concept $c$ w.r.t. $\mathcal{U}$ using $T = O(\log(m)(\log(1/\delta) + \log\log m)) \cdot \frac{1}{\beta}$ black-box oracle calls to any $(\beta, \frac{\beta}{6}, \frac{\beta}{6\ln(6)})$-barely-robust-learner $\mathbb{A}$ for $c$ w.r.t. $\mathcal{U}^{-1}(\mathcal{U})$, with total sample complexity*

$$m(\epsilon, \delta) = O\left(\frac{m_0}{\beta\epsilon}\log^2\left(\frac{m_0}{\beta\epsilon}\right) + \frac{\log(1/\delta)}{\epsilon}\right),$$

*where $m_0 = \max\left\{m_{\mathbb{A}}\left(\beta, \frac{\beta}{6}, \frac{\beta}{6\ln(6)}\right), 4\ln\left(\frac{6\ln(6)}{\beta}\right)\right\}$.*

We begin with describing the intuition behind $\beta$-RoBoost, and then we will prove Theorem 1. The proof of Corollary 2 is deferred to Appendix B.

**High-level strategy.** Let $D_c$ be the unknown distribution we want to robustly learn. Since $\mathbb{A}$ is a *barely* robust learner for $c$, calling learner $\mathbb{A}$ on an i.i.d. sample $S$ from $D_c$ will return a predictor $h_1$, where $h_1$ is robust only on a region $R_1 \subseteq \mathcal{X}$ of small mass $\beta > 0$ under distribution $D$, $\Pr_{x\sim D}[x \in R_1] \geq \beta$. We can trust the predictions of $h_1$ in the region $R_1$, but not in the complement region $\bar{R}_1$ where it is not robust. For this reason, we will use a *selective* classifier $G_{h_1}$ (see Equation 4) which makes predictions on *all* adversarial perturbations in region $R_1$, but *abstains* on adversarial perturbations not from $R_1$. In each round $t > 1$, the strategy is to focus on the region of distribution $D$ where *all* predictors $h_1, \ldots, h_{t-1}$ returned by $\mathbb{A}$ so far are *not* robust. By *rejection sampling*, $\beta$-RoBoost gives barely robust learner $\mathbb{A}$ a sample $\tilde{S}_t$ from this non-robust region, and then $\mathbb{A}$ returns a predictor $h_t$ with robustness at least $\beta$ in this region. Thus, in each round, we shrink by a factor of $\beta$ the mass of region $D$ where the predictors learned so far are not robust. After $T$ rounds, $\beta$-RoBoost outputs a *cascade* of *selective* classifiers $G_{h_1}, \ldots, G_{h_T}$ where roughly each selective classifier $G_{h_t}$ is responsible for making predictions in the region where $h_t$ is robust.

As mentioned above, one of the main components in our boosting algorithm is *selective classifiers* that essentially abstain from predicting in the region where they are *not* robust. Formally, for any classifier $h : \mathcal{X} \to \mathcal{Y}$ and any $\mathcal{U}$, denote by $G_h : \mathcal{X} \to \mathcal{Y} \cup \{\bot\}$ a *selective* classifier defined as:

$$G_h(z) \triangleq \begin{cases} y, & \text{if } (\exists y \in \mathcal{Y})(\forall \tilde{x} \in \mathcal{U}^{-1}(z)) : h(\tilde{x}) = y; \\ \bot, & \text{otherwise.} \end{cases} \tag{4}$$

Before proceeding with the proof of Theorem 1, we prove the following key Lemma about *selective classifier* $G_h$ which states that $G_h$ will *not* abstain in the region where $h$ is robust, and whenever $G_h$ predicts a label for a perturbation $z$ then we are guaranteed that this is the same label that $h$ predicts on the corresponding natural example $x$ where $z \in \mathcal{U}(x)$.

**Lemma 3.** *For any distribution $D$ over $\mathcal{X}$, any $\mathcal{U}$, and any $\beta > 0$, given a classifier $h : \mathcal{X} \to \mathcal{Y}$ satisfying $\Pr_{x\sim D}\left[x \in \mathrm{Rob}_{\mathcal{U}^{-1}(\mathcal{U})}(h)\right] \geq \beta$, then the selective classifier $G_h : \mathcal{X} \to \mathcal{Y} \cup \{\bot\}$ (see Equation 4) satisfies:*

$$\Pr_{x\sim D}\left[\forall z \in \mathcal{U}(x) : G_h(z) = h(x)\right] \geq \beta \text{ and } \Pr_{x\sim D}\left[\forall z \in \mathcal{U}(x) : G_h(z) = h(x) \vee G_h(z) = \bot\right] = 1.$$

*Proof.* Observe that, by the definition of the robust region of $h$ w.r.t. $\mathcal{U}^{-1}(\mathcal{U})$ (see Equation 3), for any $x \in \mathrm{Rob}_{\mathcal{U}^{-1}(\mathcal{U})}(h)$ the following holds:$(\forall z \in \mathcal{U}(x))(\forall \tilde{x} \in \mathcal{U}^{-1}(z)) : h(\tilde{x}) = h(x)$. By the definition of the *selective* classifier $G_h$ (see Equation 4), this implies that: $\forall z \in \mathcal{U}(x), G_h(z) = h(x)$.

Since $\Pr_{x\sim D}\left[x \in \mathrm{Rob}_{\mathcal{U}^{-1}(\mathcal{U})}(h)\right] \geq \beta$, the above implies $\Pr_{x\sim D}\left[\forall z \in \mathcal{U}(x) : G_h(z) = h(x)\right] \geq \beta$. Furthermore, for any $x \in \mathcal{X}$ and any $z \in \mathcal{U}(x)$, by definition of $\mathcal{U}^{-1}$, $x \in \mathcal{U}^{-1}(z)$. Thus, by definition of $G_h$ (see Equation 4), if $G_h(z) = y$ for some $y \in \mathcal{Y}$, then it holds that $h(x) = y$. Altogether, this implies that $\Pr_{x\sim D}\left[\forall z \in \mathcal{U}(x) : G_h(z) = h(x) \vee G_h(z) = \bot\right] = 1$. $\qquad\square$

We are now ready to proceed with the proof of Theorem 1.

*Proof of Theorem 1.* Let $\mathcal{U}$ be an arbitrary adversary, and $\mathbb{A}$ a $(\beta, \epsilon, \delta)$-barely-robust learner for some *unknown* target concept $c : \mathcal{X} \to \mathcal{Y}$ with respect to $\mathcal{U}^{-1}(\mathcal{U})$. We will show that $\beta$-RoBoost $(\epsilon, \delta)$-robustly-learns $c$ with respect to $\mathcal{U}$. Let $D$ be some unknown distribution over $\mathcal{X}$ such that $\Pr_{x \sim D} [\exists z \in \mathcal{U}(x) : c(z) \neq c(x)] = 0$. Let $\epsilon > 0$ be our target robust error, and $\epsilon'$ be the error guarantee of learner $\mathbb{A}$ (which we will set later to be $\frac{\beta\epsilon}{2}$).

Without loss of generality, suppose that $\beta$-RoBoost ran for $T = \frac{\ln(2/\epsilon)}{\beta}$ rounds (Step 4 takes care of the scenario where progress is made faster). Let $h_1 = \mathbb{A}(\tilde{S}_1), \ldots, h_T = \mathbb{A}(\tilde{S}_T)$ be the predictors returned by learner $\mathbb{A}$ on rounds $1 \leq t \leq T$. For any $1 \leq t \leq T$ and any $x \in \mathcal{X}$, denote by $R_t$ the event that $x \in \mathrm{Rob}_{\mathcal{U}^{-1}(\mathcal{U})}(h_t)$, and by $\bar{R}_t$ the event that $x \notin \mathrm{Rob}_{\mathcal{U}^{-1}(\mathcal{U})}(h_t)$ (as defined in Equation 3). Observe that by properties of $\mathbb{A}$ (see Definition 1), we are guaranteed that

$$\forall 1 \leq t \leq T : \Pr_{x \sim D_t} [R_t] \geq \beta \text{ and } \Pr_{x \sim D_t} [h_t(x) \neq c(x)] \leq \epsilon', \tag{5}$$

where $D_t$ is the distribution from which $\tilde{S}_t$ is drawn. In words, distribution $D_t$ is a conditional distribution focusing on the region of distribution $D$ where all predictors $h_1, \ldots, h_{t-1}$ are *non-robust*. Specifically, for any $x \sim D$, in case $x \in R_1 \cup \cdots \cup R_{t-1}$ (i.e., there is a predictor among $h_1, \ldots, h_{t-1}$ that is robust on $x$), then Lemma 3 guarantees that one of the selective classifiers $G_{h_1}, \ldots, G_{h_{t-1}}$ will not abstain on any $z \in \mathcal{U}(x)$: $\exists t' \leq t - 1, \forall z \in \mathcal{U}(x), G_{h_{t'}}(z) \neq \perp$. In case $x \in \bar{R}_1 \cap \cdots \cap \bar{R}_{t-1}$, then, by definition of $\bar{R}_1, \ldots, \bar{R}_{t-1}$, each of the selective classifiers $G_{h_1}, \ldots, G_{h_{t-1}}$ can be forced to abstain: $\forall t' \leq t - 1, \exists z \in \mathcal{U}(x), G_{h_{t'}}(z) = \perp$. Thus, in Step 7 of $\beta$-RoBoost, *rejection sampling* guarantees that $\tilde{S}_t$ is a sample drawn from distribution $D$ conditioned on the region $\bar{R}_{1:t-1} \triangleq \bar{R}_1 \cap \cdots \cap \bar{R}_{t-1}$. Formally, distribution $D_t$ is defined such that for any measurable event $E$:

$$\Pr_{x \sim D_t} [E] \triangleq \Pr_{x \sim D} [E | \bar{R}_1 \cap \cdots \cap \bar{R}_{t-1}]. \tag{6}$$

**Low error on natural examples.** Lemma 3 guarantees that whenever any of the *selective* classifiers $G_{h_1}, \ldots, G_{h_T}$ (see Equation 4) chooses to classify an instance $z$ at test-time their prediction will be correct with high probability. We consider two cases. First, in case of event $R_t$, $x \in \mathrm{Rob}_{\mathcal{U}^{-1}(\mathcal{U})}(h_t)$, and therefore, by Lemma 3, $\forall z \in \mathcal{U}(x) : G_{h_t}(z) = h_t(x)$. Thus, Equation 6 implies that $\forall 1 \leq t \leq T$:

$$\Pr_{x \sim D} \left[ R_t \wedge (\exists z \in \mathcal{U}(x) : (G_{h_t}(z) \neq \perp) \wedge (G_{h_t}(z) \neq c(x))) | \bar{R}_{1:t-1} \right]$$
$$= \Pr_{x \sim D_t} [R_t \wedge (\exists z \in \mathcal{U}(x) : (G_{h_t}(z) \neq \perp) \wedge (G_{h_t}(z) \neq c(x)))] \tag{7}$$
$$= \Pr_{x \sim D_t} [R_t \wedge (h_t(x) \neq c(x))].$$

Second, in case of the complement event $\bar{R}_t$, $x \notin \mathrm{Rob}_{\mathcal{U}^{-1}(\mathcal{U})}(h_t)$. Therefore, by Lemma 3, $\forall z \in \mathcal{U}(x)$, we have $G_{h_t}(z) = \perp$ or $G_{h_t}(z) = h_t(x)$. Thus,

$$\Pr_{x \sim D} \left[ \bar{R}_t \wedge (\exists z \in \mathcal{U}(x) : (G_{h_t}(z) \neq \perp) \wedge (G_{h_t}(z) \neq c(x))) | \bar{R}_{1:t-1} \right]$$
$$= \Pr_{x \sim D_t} [\bar{R}_t \wedge (\exists z \in \mathcal{U}(x) : (G_{h_t}(z) \neq \perp) \wedge (G_{h_t}(z) \neq c(x)))] \leq \Pr_{x \sim D_t} [\bar{R}_t \wedge (h_t(x) \neq c(x))]. \tag{8}$$

By law of total probability Equation 7, Equation 8, and Equation 5,

$$\Pr_{x \sim D} \left[ (R_t \vee \bar{R}_t) \wedge (\exists z \in \mathcal{U}(x) : (G_{h_t}(z) \neq \perp) \wedge (G_{h_t}(z) \neq c(x))) | \bar{R}_{1:t-1} \right]$$
$$\leq \Pr_{x \sim D_t} [R_t \wedge h_t(x) \neq c(x)] + \Pr_{x \sim D_t} [\bar{R}_t \wedge h_t(x) \neq c(x)] = \Pr_{x \sim D_t} [h_t(x) \neq c(x)] \leq \epsilon'. \tag{9}$$

**Boosted robustness.** We claim that for each $1 \leq t \leq T : \Pr_{x \sim D} [\bar{R}_{1:t}] \leq (1 - \beta)^t$. We proceed by induction on the number of rounds $1 \leq t \leq T$. In the base case, when $t = 1$, $D_1 = D$ and by Equation 5, we have $\Pr_{x \sim D} [R_1] \geq \beta$ and therefore $\Pr_{x \sim D} [\bar{R}_1] \leq 1 - \beta$.

When $t > 1$, again by Equation 5, we have that $\Pr_{x \sim D_t}[R_t] \geq \beta$ and therefore, by Equation 6, $\Pr_{x \sim D}[\bar{R}_t | \bar{R}_{1:t-1}] = \Pr_{x \sim D_t}[\bar{R}_t] \leq 1 - \beta$. Finally, by the inductive hypothesis and Bayes' rule, we get

$$\Pr_{x \sim D}[\bar{R}_{1:t}] = \Pr_{x \sim D}[\bar{R}_t | \bar{R}_{1:t-1}] \Pr_{x \sim D}[\bar{R}_{1:t-1}] \leq (1 - \beta)(1 - \beta)^{t-1} = (1 - \beta)^t. \tag{10}$$

**Analysis of robust risk.** For each $1 \leq t \leq T$, let

$$A_t = \left\{ x \in \bar{R}_t : \exists z \in \mathcal{U}(x) \text{ s.t. } G_{h_t(z)} \neq \perp \wedge \forall_{t' < t} G_{h_{t'}}(z) = \perp \right\}$$

denote the *non-robust* region of classifier $h_t$ where the *selective* classifier $G_{h_t}$ does not abstain but all selective classifiers $G_{h_1}, \ldots, G_{h_{t-1}}$ abstain. By the law of total probability, we can analyze the robust risk of the cascade predictor $\text{CAS}(h_1, \ldots, h_T)$ by partitioning the space into the *robust* and *non-robust* regions of $h_1, \ldots, h_T$. Specifically, by the structure of the cascade predictor $\text{CAS}(h_{1:T})$, each $x \sim D$ such that $\exists z \in \mathcal{U}(x)$ where $\text{CAS}(h_{1:T})(z) \neq c(x)$ satisfies the following condition:

$$\exists_{z \in \mathcal{U}(x)} : \text{CAS}(h_{1:T})(z) \neq c(x) \Rightarrow (\exists_{1 \leq t \leq T}) \left(\exists_{z \in \mathcal{U}(x)}\right) : \forall_{t' < t} G_{h_{t'}}(z) = \perp \wedge G_{h_t}(z) \neq \perp \wedge G_{h_t}(z) \neq c(x).$$

Thus, each $x \sim D$ such that $\exists z \in \mathcal{U}(x)$ where $\text{CAS}(h_{1:T})(z) \neq c(x)$ can be mapped to one (or more) of the following regions:

$$\underbrace{R_1 \vee A_1}_{G_{h_1} \text{ does not abstain}} \mid \underbrace{\bar{R}_1 \wedge (R_2 \vee A_2)}_{G_{h_2} \text{ does not abstain}} \mid \bar{R}_{1:2} \wedge (R_3 \vee A_3) \mid \ldots \mid \underbrace{\bar{R}_{1:T-1} \wedge (R_T \vee A_T)}_{G_{h_T} \text{ does not abstain}} \mid \bar{R}_{1:T}.$$

We will now analyze the robust risk based on the above decomposition:

$$\Pr_{x \sim D}[\exists z \in \mathcal{U}(x) : \text{CAS}(h_{1:T})(z) \neq c(x)]$$

$$\leq \sum_{t=1}^{T+1} \Pr_{x \sim D}\left[\bar{R}_{1:t-1} \wedge (R_t \vee A_t) \wedge (\exists z \in \mathcal{U}(x) : \text{CAS}(h_{1:T})(z) \neq c(x))\right]$$

$$\leq \sum_{t=1}^{T} \Pr_{x \sim D}\left[\bar{R}_{1:t-1} \wedge (R_t \vee A_t) \wedge (\exists z \in \mathcal{U}(x) : \text{CAS}(h_{1:T})(z) \neq c(x))\right] + \Pr_{x \sim D}[\bar{R}_{1:T}]$$

$$\overset{(i)}{\leq} \sum_{t=1}^{T} \Pr_{x \sim D}\left[\bar{R}_{1:t-1} \wedge (R_t \vee A_t) \wedge (\exists z \in \mathcal{U}(x) : G_{h_t}(z) \neq \perp \wedge G_{h_t}(z) \neq c(x))\right] + \Pr_{x \sim D}[\bar{R}_{1:T}]$$

$$= \sum_{t=1}^{T} \Pr_{x \sim D}[\bar{R}_{1:t-1}] \Pr_{x \sim D}\left[(R_t \vee A_t) \wedge (\exists z \in \mathcal{U}(x) : G_{h_t}(z) \neq \perp \wedge G_{h_t}(z) \neq c(x)) | \bar{R}_{1:t-1}\right] + \Pr_{x \sim D}[\bar{R}_{1:T}]$$

$$\overset{(ii)}{\leq} \sum_{t=1}^{T} (1 - \beta)^{t-1} \epsilon' + (1 - \beta)^T = \epsilon' \sum_{t=1}^{T} (1 - \beta)^{t-1} = \epsilon' \frac{1 - (1 - \beta)^T}{1 - (1 - \beta)} + (1 - \beta)^T \leq \frac{\epsilon'}{\beta} + (1 - \beta)^T, \tag{11}$$

where inequality $(i)$ follows from the definitions of $\bar{R}_{1:t-1}$, $R_t$, and $A_t$, and inequality $(ii)$ follows from Equation 10 and Equation 9. It remains to choose $T$ and $\epsilon'$ such that the robust risk is at most $\epsilon$.

**Sample and oracle complexity.** It suffices to choose $T = \frac{\ln(2/\epsilon)}{\beta}$ and $\epsilon' = \frac{\epsilon \beta}{2}$. We next analyze the sample complexity. Fix an arbitrary round $1 \leq t \leq T$. In order to obtain a *good* predictor $h_t$ from learner $\mathbb{A}$ satisfying Equation 5, we need to draw $m_{\mathbb{A}}(\beta, \beta \epsilon/2, \delta/2T)$ samples from $D_t$. We do this by drawing samples from the original distribution $D$ and doing rejection sampling. Specifically, let $m = \max\left\{ m_{\mathbb{A}}(\beta, \frac{\epsilon \beta}{2}, \frac{\delta}{2T}), 4 \ln\left(\frac{2T}{\delta}\right) \right\}$ (as defined in Step 1). Then, for each $1 \leq i \leq m$, let $X_{t,i}$ be a random variable counting the number of samples $(x, y)$ drawn from $D$ until a sample $(x, y) \in \bar{R}_{1:t-1}$ is obtained. Notice that $X_{t,i}$ is a geometric random variable with expectation $1/p_t$ where $p_t = \Pr_{x \sim D}[\bar{R}_{1:t}]$. Then, the expected number of samples drawn from $D$ in round $t$ is $\mathbb{E}[\sum_{i=1}^{m} X_{t,i}] = \frac{m}{p_t}$. By applying a standard concentration inequality for the sums of i.i.d. geometric random variables [4], we get

$$\Pr\left[\sum_{i=1}^{m} X_{t,i} > 2\frac{m}{p_t}\right] \leq e^{-\frac{2m(1 - 1/2)^2}{2}} = e^{-\frac{m}{4}} \leq \frac{\delta}{2T},$$

where the last inequality follows from our choice of $m$. By a standard union bound, we get that with probability at least $1 - \frac{\delta}{2}$, the total number of samples $\sum_{t=1}^{T} \sum_{i=1}^{m} X_{t,i} \leq \sum_{t=1}^{T} 2\frac{m}{p_t} = 2m \sum_{t=1}^{T} \frac{1}{p_t} \leq \frac{4mT}{\epsilon}$. $\qquad\square$

## 4 The Necessity of barely robust learning

We have established in Section 3 (Theorem 1) that our proposed notion of barely robust learning in Definition 1 *suffices* for strongly robust learning. But is our notion actually *necessary* for strongly robust learning? In particular, notice that in our proposed notion of barely robust learning in Definition 1, we require $\beta$-robustness with respect to a "larger" perturbation set $\mathcal{U}^{-1}(\mathcal{U})$, instead of the actual perturbation set $\mathcal{U}$ that we care about. Is this necessary? or can we perhaps boost robustness even with the weaker guarantee of $\beta$-robustness with respect to $\mathcal{U}$?

In this section, we answer this question in the negative. First, we provably show in Theorem 4 that strongly robust learning with respect to $\mathcal{U}$ implies barely robust learning with respect to $\mathcal{U}^{-1}(\mathcal{U})$. This indicates that our proposed notion of barely robust learning with respect to $\mathcal{U}^{-1}(\mathcal{U})$ (Definition 1) is *necessary* for strongly robust learning with respect to $\mathcal{U}$. Second, we provably show in Theorem 6 that barely robust learning with respect to $\mathcal{U}$ does *not* imply strongly robust learning with respect to $\mathcal{U}$ when the robustness parameter $\beta \leq \frac{1}{2}$ which is the main regime of interest that we study in this work.

**Theorem 4.** *For any $\mathcal{U}$, learner $\mathbb{B}$, and $\epsilon \in (0, 1/4)$, if $\mathbb{B}$ $(\epsilon, \delta)$-robustly learns some unknown target concept $c$ w.r.t. $\mathcal{U}$, then there is a learner $\tilde{\mathbb{B}}$ that $(\frac{1-\epsilon}{2}, 2\epsilon, 2\delta)$-barely-robustly-learns $c$ w.r.t. $\mathcal{U}^{-1}(\mathcal{U})$.*

We defer the proof of Theorem 4 to Appendix C, but we briefly describe the high-level strategy here. The main idea is to convert a strongly robust learner $\mathbb{B}$ with respect to $\mathcal{U}$ to a barely robust learner $\tilde{\mathbb{B}}$ with respect to $\mathcal{U}^{-1}(\mathcal{U})$. We do this with a simple *expansion* trick that modifies a predictor $h$ robust with respect to $\mathcal{U}$ to a predictor $g$ robust with respect to $\mathcal{U}^{-1}(\mathcal{U})$. For each label $y \in \mathcal{Y}$, we do this expansion conditional on the label to get a predictor $g_y$ that is robust w.r.t. $\mathcal{U}^{-1}(\mathcal{U})$ but only in the region of $\mathcal{X}$ where $h$ predicts the label $y$ robustly w.r.t. $\mathcal{U}$. This is described in the following key Lemma. We then use fresh samples to select predictor a $g_y$ whose label $y$ occurs more often.

**Lemma 5.** *For any distribution $D$ over $\mathcal{X}$ and any concept $c : \mathcal{X} \to \{\pm 1\}$, given a predictor $\hat{h} : \mathcal{X} \to \{\pm 1\}$ such that $\mathrm{R}_{\mathcal{U}}(\hat{h}; D_c) \leq \epsilon$ for some $\epsilon \in (0, 1/4)$, then for each $y \in \{\pm 1\}$, the predictor $g_y$ defined for each $x \in \mathcal{X}$ as*

$$g_y(x) \triangleq y \text{ iff } x \in \bigcup_{\tilde{x} \in \mathrm{Rob}_{\mathcal{U}}(\hat{h}) \wedge \hat{h}(\tilde{x}) = y} \mathcal{U}^{-1}(\mathcal{U}(\tilde{x})) \quad \textit{satisfies}$$

$$\Pr_{x \sim D}[g_y(x) \neq c(x)] \leq 2\epsilon \text{ and } \Pr_{x \sim D}\left[x \in \mathrm{Rob}_{\mathcal{U}^{-1}(\mathcal{U})}(g_y)\right] \geq (1-\epsilon) \Pr_{x \sim D}\left[\hat{h}(x) = y \big| x \in \mathrm{Rob}_{\mathcal{U}}(\hat{h})\right].$$

As mentioned earlier, Theorem 4 still leaves open the question of whether the *weaker* requirement of barely robust learning with respect to $\mathcal{U}$ suffices for strongly robust learning with respect to $\mathcal{U}$. We show next that this weaker requirement is *not* sufficient.

**Theorem 6.** *There is an instance space $\mathcal{X}$, a perturbation set $\mathcal{U}$, and a class $\mathcal{C}$ such that $\mathcal{C}$ is $(\beta = \frac{1}{2}, \epsilon = 0, \delta)$-barely-robustly-learnable with respect to $\mathcal{U}$, but $\mathcal{C}$ is not $(\epsilon, \delta)$-robustly-learnable with respect to $\mathcal{U}$ for any $\epsilon < 1/2$.*

We defer the proof of Theorem 6 to Appendix D, but we briefly sketch the high-level argument here. To show this impossibility result, we construct a collection of distributions that is *barely* robustly learnable w.r.t. $\mathcal{U}$ with robustness parameter $\beta = \frac{1}{2}$ and natural error $\epsilon = 0$ using a randomized predictor. We also show that it is not possible to robustly learn this collection with robust risk strictly smaller than $\frac{1}{2}$. The second part is shown by relying on a necessary condition for strongly robust learning proposed by [15] which is the finiteness of the robust shattering dimension.

## 5 Discussion

In this paper, we put forward a theory for *boosting* adversarial robustness. We discuss below practical implications and outstanding directions that remain to be addressed.

**Practical implications.** Our algorithm $\beta$-RoBoost is generic and can be used with any black-box barely robust learner $\mathbb{A}$. In the context of deep learning and $\ell_p$ robustness, our results suggest the following: for targeted robustness of radius $\gamma$, use an adversarial learning method [e.g., 13, 21, 6] to learn a neural net $h_{\mathrm{NN}}^1$ predictor robust with radius $2\gamma$, then filter the training examples to include *only* the ones on which $h_{\mathrm{NN}}^1$ is *not* robust with radius $2\gamma$, and repeat the training process on the filtered examples to learn a second neural net, and so on. Finally, use the cascade of neural nets $\mathrm{CAS}(h_{\mathrm{NN}}^1, h_{\mathrm{NN}}^2, \ldots)$ as defined in $\beta$-RoBoost to predict.

It would be interesting to empirically explore whether adversarial learning methods [e.g., 13, 21, 6] satisfy the *barely* robust learning condition: on each round of boosting, the learning algorithm can shrink the fraction of the training examples on which the predictor from the previous round is not robust on. This is crucial for progress.

**Multiclass.** We would like to emphasize that our theory for boosting *barely robust* learners extends seamlessly to multiclass learning problems. In particular, when the label space $|\mathcal{Y}| > 2$, we obtain the same guarantees in Theorem 1 using the same algorithm $\beta$-RoBoost. The other direction of converting an $(\epsilon, \delta)$-robust-learner with respect to $\mathcal{U}$ to an $(\beta, 2\epsilon, 2\delta)$-barely-robust-learner with respect to $\mathcal{U}^{-1}(\mathcal{U})$ also holds, using the same technique in Theorem 4, but now we get $\beta = \frac{1-\epsilon}{|\mathcal{Y}|}$.

**Relationship between robustness parameter $\beta$ and error parameter $\epsilon$.** To achieve robust risk at most $\epsilon$ using a barely robust learner $\mathbb{A}$ with robustness parameter $\beta$, our algorithm $\beta$-RoBoost requires $\mathbb{A}$ to achieve a natural error of $\tilde{\epsilon} = \frac{\beta \epsilon_0}{2}$ for any constant $\epsilon_0 < \frac{1}{2}$ (say $\tilde{\epsilon} = \frac{\beta}{6}$) (see Corollary 2). It would be interesting to resolve whether requiring natural risk $\tilde{\epsilon}$ that depends on $\beta$ is necessary, or whether it is possible to avoid dependence on $\beta$. Concretely, an open question here is: can we achieve robust risk at most $\epsilon$ using a $(\beta, O(\epsilon), \delta)$-barely-robust-learner instead of requiring $(\beta, O(\beta), \delta)$-barely-robust-learner? It actually suffices to answer the following: given a $(\beta, \frac{1}{3}, \delta)$-barely-robust-learner, is it possible to achieve *robust* risk at most $\frac{1}{3}$? A related question is whether it is possible to boost the error while maintaining robustness fixed at some level. For example, given a $(\beta, \frac{1}{3}, \delta)$-barely-robust-learner $\mathbb{A}$, is it possible to boost this to a $(\beta, \epsilon, \delta)$-barely-robust-learner $\mathbb{B}$?

**Agnostic setting.** We focused only on boosting robustness in the *realizable* setting, where the target unknown concept $c$ and the unknown distribution $D$ satisfy $\mathrm{R}_{\mathcal{U}}(c; D_c) = 0$, and we showed an equivalence between barely-robust-learning and strongly-robust-learning in this setting. In terms of obtaining stronger guarantees, we remark that an agnostic-to-realizable reduction described in [15, Theorem 8], implies, together with Theorem 1, that barely-robust-learning is equivalent to agnostic-robust-learning. The agnostic-to-realizable reduction in [15] is not oracle-efficient, however, so the interesting open question here is to explore oracle-efficient boosting algorithms in the agnostic setting. A key step in this direction is to identify reasonable definitions of *agnostic*-barely-robust learners that go beyond Definition 1.

**Computational efficiency.** Our algorithm $\beta$-RoBoost is guaranteed to terminate in at most $T = \frac{\ln(2/\epsilon)}{\beta}$ rounds of boosting (see Theorem 1), because our rejection sampling mechanism (Step 8) will safely terminate when the mass of the non-robust region becomes small (mass at most $\epsilon/2$). In each round $t > 1$ of boosting, rejection sampling (Step 7) in $\beta$-RoBoost needs to sample from the region of the distribution where all previous predictors $h_1, \ldots, h_{t-1}$ are *non-robust*, and doing this computationally efficiently requires an efficient procedure to *certify* robustness of $h_i$ on perturbation set $\mathcal{U}^{-1}(\mathcal{U})(x)$. Certifying robustness of predictors is an active area of research [e.g., randomized smoothing with $\ell_2$ perturbations in 6], but is outside the scope of this work, since we make no assumptions on the form of the returned predictors $h_i$ or the perturbation sets $\mathcal{U}$ and $\mathcal{U}^{-1}(\mathcal{U})$. In general, it's hard to imagine a world where efficient robust learning is possible without efficient certification of robustness. In other words, efficient certification of robustness seems to be a "necessary" condition for efficient robust learning, not just for our proposed boosting algorithm, but for any algorithm that is devised for robust learning. Without efficient certification of robustness, we can't even compute the robust loss, so how can we even learn.

**Comparing $\mathcal{U}^{-1}(\mathcal{U})$ vs. $\mathcal{U}$.** As mentioned earlier in the introduction, if $\mathcal{U}$ is a metric-ball of radius $\gamma$, then $\mathcal{U}^{-1}(\mathcal{U})$ is a metric-ball of radius $2\gamma$. More generally, we think of robustness w.r.t. $\mathcal{U}^{-1}(\mathcal{U})$ as being robust to "twice" the adversary's power (which is represented by $\mathcal{U}$). For example, for a

general vector space $\mathcal{X}$, if $\mathcal{U}(x) = x + B$, where $B$ is some arbitrary symmetric set (i.e. $v \in B$ if and only if $-v \in B$), then $\mathcal{U}^{-1}(\mathcal{U})(x) = x + B + B$. We do not yet know of examples of $\mathcal{U}$ where $\mathcal{U}^{-1}(\mathcal{U})$ is much "larger" than $\mathcal{U}$, but this is an interesting direction to explore. More generally, it would be interesting to explore whether barely-robust-learning w.r.t. $\mathcal{U}^{-1}(\mathcal{U})$ is *computationally* equivalent to strong-robust-learning w.r.t. $\mathcal{U}$.

## 6    Simple Experiments

We conduct simple experiments that illustrate the utility of our theoretical contribution for boosting robustness. These experiments demonstrate that our algorithm, $\beta$-RoBoost, can boost and improve the robustness of black-box learning algorithms. We describe the setup and the results below.

**Datasets.**    A synthetic binary classification dataset (`make_moons` from scikit-learn), and `MNIST` (rescaled by dividing by 255, and converted to binary classification of odd vs. even).

**Perturbation set $\mathcal{U}$.**    We consider $\ell_2$ perturbations of some radius $\gamma$. In the respective datasets, we computed the minimum distance between examples from different classes and chose a radius $\gamma$ that's smaller than this minimum distance.

**Barely Robust Learner.**    We use an off-the-shelve Linear SVM solver (from scikit-learn) as a barely-robust-learner. Using linear predictors and $\ell_2$ perturbations simplifies the computation of the robust loss since it exactly corresponds to computing the margin loss [see Lemma 4.2 in 14]. Formally, for any linear predictor $w$ and example $(x, y)$,

$$\sup_{\|\delta\|_2 \leq \gamma} \mathbb{1}[\text{sign}(\langle w, x + \delta \rangle) \neq y] = \mathbb{1}\left[ y \left\langle \frac{w}{\|w\|_2}, x \right\rangle \leq \gamma \right].$$

We ran our boosting algorithm, $\beta$-RoBoost, and compared it against the baseline of a single Linear SVM call. In our boosting algorithm, we run for as many rounds as possible until there are no more examples left in the training set to run LinearSVM on.

**Results**    On `make_moons` with perturbation radius $\gamma = 0.1$, the baseline Linear SVM achieves a robust accuracy of 84.78%, while $\beta$-RoBoost (with 2 rounds of boosting) achieves robust accuracy of 89.86%. On `MNIST` with perturbation radius $\gamma = 0.5$, the baseline Linear SVM achieves a robust accuracy of 73.9%, while $\beta$-RoBoost (with 2 rounds of boosting) achieves robust accuracy of 80.05%. Finally, on `MNIST` with a bigger perturbation radius $\gamma = 1.0$, the baseline Linear SVM achieves a robust accuracy of 48.1%, while $\beta$-RoBoost (with 2 rounds of boosting) achieves robust accuracy of 70.12%.

We observe that $\beta$-RoBoost improves the robustness of Linear SVM. Notice that even in the regime where the baseline Linear SVM archives robust-accuracy $< 50\%$ (MNIST with perturbation radius 1.0), $\beta$-RoBoost can actually improve the robust-accuracy beyond 50%.

We include code to reproduce our `MNIST` experiments with perturbation radius $\gamma = 1.0$ in Appendix F.

## Acknowledgments and Disclosure of Funding

We would like to thank Steve Hanneke and Nathan Srebro for insightful discussions, and the anonymous reviewers for their detailed and valuable feedback. This work was supported in part by DARPA under cooperative agreement HR00112020003.[1] This work was supported in part by the National Science Foundation under grant CCF-1815011.

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
