## A Boosting Robustness with Unlabeled Data

Prior work has shown that unlabeled data can improve adversarially robust generalization in practice [2, 5], and there is also theoretical work quantifying the benefit of unlabeled data for robust generalization [3]. In this section, we highlight yet another benefit of unlabeled data for adversarially robust learning. Specifically, we show that it is *possible* to boost robustness by relying only on *unlabeled* data.

We will start with some intuition first. For an unknown distribution $D_c$, imagine having access to a *non-robust* classifier $h$ that makes no mistakes on natural examples, i.e., $\Pr_{x \sim D}[h(x) \neq c(x)] = 0$ but $\Pr_{x \sim D}[\exists z \in \mathcal{U}(x) : h(z) \neq h(x)] = 1$. Now, in order to learn a robust classifier, we can use $\beta$-RoBoost where in each round of boosting we sample unlabeled data from $D$ (label it with $h$) and call a barely robust learner $\mathbb{A}$ on this *pseudo-labeled* dataset.

This highlights that perhaps robustness can be boosted using only unlabeled data if we have access to a good *pseudo-labeler* $h$ that makes few mistakes on natural examples from $D$. But in case that $\Pr_{x \sim D}[h(x) \neq c(x)] = \epsilon$ for some small $\epsilon > 0$, it no longer suffices to use a barely robust learner $\mathbb{A}$ for $c$, but rather we need a more powerful learner that is tolerant to the noise in the labels introduced by $h$. Formally, we introduce the following noise-tolerant barely robust learner:

**Definition 3** (Noise-Tolerant Barely Robust Learner). Learner $\mathbb{A}$ $(\eta, \beta, \epsilon, \delta)$-barely-robustly-learns a concept $c : \mathcal{X} \to \mathcal{Y}$ w.r.t. $\mathcal{U}^{-1}(\mathcal{U})$ if there exists $m(\eta, \beta, \epsilon, \delta) \in \mathbb{N}$ such that for any distribution $D$ over $\mathcal{X}$ satisfying $\Pr_{x \sim D}[\exists z \in \mathcal{U}(x) : c(z) \neq c(x)] = 0$ and any $h : \mathcal{X} \to \mathcal{Y}$ where $\Pr_{x \sim D}[h(x) \neq c(x)] \leq \eta$, w.p. at least $1 - \delta$ over $S \sim D_h^m$, $\mathbb{A}$ outputs a predictor $\hat{h} = \mathbb{A}(S)$ satisfying:

$$\Pr_{x \sim D}\left[x \in \mathrm{Rob}_{\mathcal{U}^{-1}(\mathcal{U})}(\hat{h})\right] \geq \beta \quad \text{and} \quad \Pr_{x \sim D}\left[\hat{h}(x) \neq c(x)\right] \leq \Pr_{x \sim D}[h(x) \neq c(x)] + \epsilon \leq \eta + \epsilon.$$

In Theorem 7, we show that given a *noise-tolerant* barely robust learner $\mathbb{A}$ for some unknown target concept $c$ (according to Definition 3), it is *possible* to strongly robustly learn $c$ with $\beta$-URoBoost by making black-box oracle calls to $\mathbb{A}$.

---
**Algorithm 2:** $\beta$-URoBoost — Boosting Robustness with Unlabeled Data

---
**Input:** Sampling oracle for distribution $D_c$, black-box noise-tolerant barely-robust learner $\mathbb{A}$.
1  Draw $m = m_{\mathbb{A}}(\eta, \beta, \frac{\beta\epsilon}{4}, \frac{\delta}{2})$ labeled samples $S = \{(x_1, y_1), \dots, (x_m, y_m)\} \sim D_c$.
2  Call learner $\mathbb{A}$ on $S$ and let predictor $\hat{h} = \mathbb{A}(S)$ be its output.
3  Call $\beta$-RoBoost with access to labeled samples from $D_{\hat{h}}$ (i.e., $(x, \hat{h}(x)) \sim D_{\hat{h}}$), and
   black-box $(\eta, \beta, \frac{\epsilon\beta}{4}, \frac{\delta}{2T})$-noise-tolerant-barely-robust-learner $\mathbb{A}$.
**Output:** The cascade predictor $\mathrm{CAS}(h_1, \dots, h_T)$.

---

**Theorem 7.** *For any perturbation set $\mathcal{U}$, $\beta$-URoBoost $(\epsilon, \delta)$-robustly-learns any target concept $c$ w.r.t. $\mathcal{U}$ using $T + 1 \leq \frac{\ln(2/\epsilon)}{\beta} + 1$ black-box oracle calls to any $(\eta, \beta, \frac{\beta\epsilon}{4}, \frac{\delta}{2T})$-barely-robust learner $\mathbb{A}$ for $c$ w.r.t. $\mathcal{U}^{-1}(\mathcal{U})$, with* labeled *sample complexity of $m_{\mathbb{A}}(\eta, \beta, \frac{\beta\epsilon}{4}, \frac{\delta}{2})$ and* unlabeled *sample complexity of at most*

$$\frac{4T \max\left\{m_{\mathbb{A}}(\eta, \beta, \frac{\beta\epsilon}{4}, \frac{\delta}{4T}), 4\ln\left(\frac{4T}{\delta}\right)\right\}}{\epsilon}.$$

*Proof.* Let $\mathcal{U}$ be an arbitrary perturbation set, and $\mathbb{A}$ a $(\eta, \beta, \epsilon, \delta)$-barely-robust learner for some *unknown* target concept $c : \mathcal{X} \to \mathcal{Y}$ with respect to $\mathcal{U}^{-1}(\mathcal{U})$. We will show that $\beta$-URoBoost $(\epsilon, \delta)$-robustly-learns $c$ with respect to $\mathcal{U}$. Let $D$ be some unknown distribution over $\mathcal{X}$ such that $\Pr_{x \sim D}[\exists z \in \mathcal{U}(x) : c(z) \neq c(x)] = 0$.

By Step 1 and Step 2 and the guarantee of learner $\mathbb{A}$ (see Definition 3), with probability at least $1 - \frac{\delta}{2}$ over $S^m \sim D_c$, it holds that

$$\Pr_{x \sim D}\left[\hat{h}(x) \neq c(x)\right] \leq \frac{\beta\epsilon}{4}.$$

That is, with high probability, $\hat{h}$ is a predictor with low error on *natural* examples.

**Pseudo labeling.** In Step 3, $\beta$-URoBoost essentially runs $\beta$-RoBoost using unlabeled samples from $D$ that are labeled with the predictor $\hat{h}$. Thus, we can view this as robustly learning the concept $\hat{h}$ which is only an approximation of the true concept $c$ that we care about. Since the noise tolerance $\eta \geq \frac{\beta\epsilon}{4}$, it follows by the guarantees of learner $\mathbb{A}$ (see Definition 3) and Equation 11, that

$$\Pr_{x \sim D} \left[ \exists z \in \mathcal{U}(x) : \text{CAS}(h_{1:T})(z) \neq c(x) \right] \leq \frac{\Pr_{x \sim D}\left[\hat{h}(x) \neq c(x)\right]}{\beta} + \frac{\beta\epsilon}{4\beta} + (1-\beta)^T \leq \frac{\epsilon}{4} + \frac{\epsilon}{4} + \frac{\epsilon}{2} \leq \epsilon.$$

$\square$

# B   Auxiliary Lemmas and Proofs for Corollary 2

---
**Algorithm 3:** $\alpha$-Boost — Boosting *weakly* robust learners

**Input:** Training dataset $S = \{(x_1, y_1), \ldots, (x_m, y_m)\}$, black-box *weak* robust learner $\mathbb{B}$.

1  Set $m_0 = m_{\mathbb{B}}(1/3, 1/3)$.
2  Initialize $D_1$ to be uniform over $S$, and set $T = O(\log m)$.
3  **for** $1 \leq t \leq T$ **do**
4      Sample $S_t \sim D_t^{m_0}$, call learner $\mathbb{B}$ on $S_t$, and denote by $h_t$ its output predictor. Repeat this step until $\text{R}_{\mathcal{U}}(h_t; D_t) \leq 1/3$.
5      Compute a new distribution $D_{t+1}$ by applying the following update for each $(x, y) \in S$:

$$D_{t+1}(\{(x, y)\}) = \frac{D_t(\{(x, y)\})}{Z_t} \times \begin{cases} e^{-2\alpha}, & \text{if } \mathbb{1}[\forall z \in \mathcal{U}(x) : h_{t-1}(z) = y] = 1; \\ 1, & \text{otherwise}, \end{cases}$$

    where $Z_t$ is a normalization factor and $\alpha = 1/8$.

**Output:** A majority-vote classifier $\text{MAJ}(h_1, \ldots, h_T)$.

---

Before proceeding with the proof of Corollary 2, we state the following guarantee that $\alpha$-Boost provides for boosting weakly robust learners.

**Lemma 8** ([15])**.** *For any perturbation set $\mathcal{U}$, $\alpha$-Boost $(\epsilon, \delta)$-robustly-learns any target concept $c$ w.r.t. $\mathcal{U}$ using $T$ black-box oracle calls to any $(\frac{1}{3}, \frac{1}{3})$-robust-learner $\mathbb{B}$ for $c$ w.r.t. $\mathcal{U}$, with total sample complexity*

$$m(\epsilon, \delta) = O\left( \frac{m_{\mathbb{B}}(1/3, 1/3)}{\epsilon} \log^2\left( \frac{m_{\mathbb{B}}(1/3, 1/3)}{\epsilon} \right) + \frac{\log(1/\delta)}{\epsilon} \right),$$

*and oracle calls $T = O(\log(m)(\log(1/\delta) + \log\log m))$.*

**Lemma 9** (Sample Compression Robust Generalization Guarantee – [15])**.** *For any $k \in \mathbb{N}$ and fixed function $\phi : (\mathcal{X} \times \mathcal{Y})^k \to \mathcal{Y}^{\mathcal{X}}$, for any distribution $P$ over $\mathcal{X} \times \mathcal{Y}$ and any $m \in \mathbb{N}$, for $S = \{(x_1, y_1), \ldots, (x_m, y_m)\}$ i.i.d. $P$-distributed random variables, with probability at least $1 - \delta$, if $\exists i_1, \ldots, i_k \in \{1, \ldots, m\}$ s.t. $\hat{R}_{\mathcal{U}}(\phi((x_{i_1}, y_{i_1}), \ldots, (x_{i_k}, y_{i_k})); S) = 0$, then*

$$\text{R}_{\mathcal{U}}(\phi((x_{i_1}, y_{i_1}), \ldots, (x_{i_k}, y_{i_k})); P) \leq \frac{1}{m - k}(k \ln(m) + \ln(1/\delta)).$$

*Proof of Lemma 8.* Let $\mathbb{B}$ be a *weak* robust learner with fixed parameters $(\epsilon_0, \delta_0) = (1/3, 1/3)$ for some *unknown* target concept $c$ with respect to $\mathcal{U}$. Let $D$ be some unknown distribution over $\mathcal{X}$ such that $\Pr_{x \sim D}\left[\exists z \in \mathcal{U}(x) : c(z) \neq c(x)\right] = 0$. By Definition 2, with fixed sample complexity $m_0 = m_{\mathbb{B}}(1/3, 1/3)$, for any distribution $\tilde{D}$ over $\mathcal{X}$ such that $\Pr_{x \sim \tilde{D}}\left[\exists z \in \mathcal{U}(x) : c(z) \neq c(x)\right] = 0$, with probability at least $1/3$ over $S \sim \tilde{D}_c^{m_0}$, $\text{R}_{\mathcal{U}}(\mathbb{B}(S); \tilde{D}_c) \leq 1/3$.

We will now boost the confidence and robust error guarantee of the *weak* robust learner $\mathbb{B}$ by running boosting with respect to the *robust* loss (rather than the standard 0-1 loss). Specifically, fix $(\epsilon, \delta) \in (0, 1)$ and a sample size $m(\epsilon, \delta)$ that will be determined later. Let $S = \{(x_1, y_1), \ldots, (x_m, y_m)\}$ be an i.i.d. sample from $D_c$. Run the $\alpha$-Boost algorithm on dataset $S$ using $\mathbb{B}$ as the weak robust learner

for a number of rounds $T$ that will be determined below. On each round $t$, $\alpha$-Boost computes an empirical distribution $D_t$ over $S$ by applying the following update for each $(x, y) \in S$:

$$D_t(\{(x, y)\}) = \frac{D_{t-1}(\{(x, y)\})}{Z_{t-1}} \times \begin{cases} e^{-2\alpha}, & \text{if } \mathbb{1}[\forall z \in \mathcal{U}(x) : h_{t-1}(z) = y] = 1; \\ 1, & \text{otherwise,} \end{cases}$$

where $Z_{t-1}$ is a normalization factor, $\alpha$ is a parameter that will be determined below, and $h_{t-1}$ is the *weak* robust predictor outputted by $\mathbb{B}$ on round $t-1$ that satisfies $\mathrm{R}_{\mathcal{U}}(h_{t-1}; D_{t-1}) \leq 1/3$. Once $D_t$ is computed, we sample $m_0$ examples from $D_t$ and run *weak* robust learner $\mathbb{B}$ on these examples to produce a hypothesis $h_t$ with robust error guarantee $\mathrm{R}_{\mathcal{U}}(h_t; D_t) \leq 1/3$. This step has failure probability at most $\delta_0 = 1/3$. We will repeat it for at most $\lceil \log(2T/\delta) \rceil$ times, until $\mathbb{B}$ succeeds in finding $h_t$ with robust error guarantee $\mathrm{R}_{\mathcal{U}}(h_t; D_t) \leq 1/3$. By a union bound argument, we are guaranteed that with probability at least $1 - \delta/2$, for each $1 \leq t \leq T$, $\mathrm{R}_{\mathcal{U}}(h_t; D_t) \leq 1/3$. Following the argument from [18, Section 6.4.2], after $T$ rounds we are guaranteed

$$\min_{(x,y) \in S} \frac{1}{T} \sum_{t=1}^{T} \mathbb{1}[\forall z \in \mathcal{U}(x) : h_t(z) = y] \geq \frac{2}{3} - \frac{2}{3}\alpha - \frac{\ln(|S|)}{2\alpha T},$$

so we will plan on running until round $T = 1 + 48 \ln(|S|)$ with value $\alpha = 1/8$ to guarantee

$$\min_{(x,y) \in S} \frac{1}{T} \sum_{t=1}^{T} \mathbb{1}[\forall z \in \mathcal{U}(x) : h_t(z) = y] > \frac{1}{2},$$

so that the majority-vote classifier $\mathrm{MAJ}(h_1, \ldots, h_T)$ achieves *zero* robust loss on the empirical dataset $S$, $\mathrm{R}_{\mathcal{U}}(\mathrm{MAJ}(h_1, \ldots, h_L); S) = 0$.

Note that each of these classifiers $h_t$ is equal to $\mathbb{B}(S'_t)$ for some $S'_t \subseteq S$ with $|S'_t| = m_0$. Thus, the classifier $\mathrm{MAJ}(h_1, \ldots, h_T)$ is representable as the value of an (order-dependent) reconstruction function $\phi$ with a compression set size $m_0 T = m_0 O(\log m)$. Now, invoking Lemma 9, with probability at least $1 - \delta/2$,

$$\mathrm{R}_{\mathcal{U}}(\mathrm{MAJ}(h_1, \ldots, h_T); \mathcal{D}) \leq O\left(\frac{m_0 \log^2 m}{m} + \frac{\log(2/\delta)}{m}\right),$$

and setting this less than $\epsilon$ and solving for a sufficient size of $m$ yields the stated sample complexity bound. $\qquad \square$

We are now ready to proceed with the proof of Corollary 2.

*Proof of Corollary 2.* The main idea is to perform two layers of boosting. In the first layer, we use $\beta$-RoBoost to get a *weak* robust learner for $c$ w.r.t. $\mathcal{U}$ from a *barely* robust learner $\mathbb{A}$ for $c$ w.r.t. $\mathcal{U}^{-1}(\mathcal{U})$. Then, in the second layer, we use $\alpha$-Boost to boost $\beta$-RoBoost from a *weak* robust learner to a *strong* robust learner for $c$ w.r.t. $\mathcal{U}$.

Let $\mathbb{A}$ be a $(\beta, \frac{\beta}{6}, \frac{\beta}{6\ln(6)})$-barely-robust-learner $\mathbb{A}$ for $c$ w.r.t. $\mathcal{U}^{-1}(\mathcal{U})$. Let $(\epsilon_0, \delta_0, T_0) = (\frac{1}{3}, \frac{1}{3}, \frac{\ln(6)}{\beta})$, and observe that $\mathbb{A}$ is a $(\beta, \frac{\beta\epsilon_0}{2}, \frac{\delta_0}{2T_0})$-barely-robust-learner for $c$ w.r.t. $\mathcal{U}^{-1}(\mathcal{U})$. By Theorem 1, $\beta$-RoBoost $(\epsilon_0, \delta_0)$-robustly-learns $c$ w.r.t. $\mathcal{U}$ using $T_0$ black-box oracle calls to $\mathbb{A}$, with sample complexity $m_0 = O\left(\frac{\max\{m_{\mathbb{A}}, 4\ln\left(\frac{6\ln(6)}{\beta}\right)\}}{\beta}\right)$. Finally, by Lemma 8, $\alpha$-Boost $(\epsilon, \delta)$-robustly-learns $c$ w.r.t. $\mathcal{U}$ using $m(\epsilon, \delta) = O\left(\frac{m_0}{\epsilon} \log\left(\frac{m_0}{\epsilon}\right) + \frac{\log(1/\delta)}{\epsilon}\right)$ samples, and $O(\log m)$ black-box oracle calls to $\beta$-RoBoost. $\qquad \square$

# C Proof of Theorem 4

*Proof of Lemma 5.* Without loss of generality, let $y = +1$. Let $x \in \mathrm{supp}(D)$ such that $x \in \mathrm{Rob}_{\mathcal{U}}(\hat{h})$. In case $\hat{h}(x) = +1$, then by definition of $g_+$, since $x \in \mathcal{U}^{-1}(\mathcal{U}(x))$, it holds that $g_+(x) = +1$. In case $\hat{h}(x) = -1$, then $\neg \exists \tilde{x} \in \mathrm{Rob}_{\mathcal{U}}(\hat{h})$ such that $\hat{h}(\tilde{x}) = +1$ and $\mathcal{U}(\tilde{x}) \cap \mathcal{U}(x) \neq \emptyset$, which implies that $x \notin \bigcup_{\tilde{x} \in \mathrm{Rob}_{\mathcal{U}}(\hat{h}) \wedge \hat{h}(\tilde{x}) = +1} \mathcal{U}^{-1}(\mathcal{U}(\tilde{x}))$, and therefore, $g_+(x) = -1$. This establishes that in the

robust region of $\hat{h}$, $\mathrm{Rob}_{\mathcal{U}}(\hat{h})$, the predictions of $g_+$ on *natural* examples $x \sim D$ are equal to the predictions of $\hat{h}$. We will use this observation, in addition to the fact that the robust risk of $\hat{h}$ is small $\left(\mathrm{Pr}_{x \sim D}\left[\exists z \in \mathcal{U}(x) : \hat{h}(z) \neq c(x)\right] \leq \epsilon\right)$ to show that the error of $g_+$ on natural examples is small. Specifically, by law of total probability,

$$
\begin{aligned}
\Pr_{x \sim D}\left[g_+(x) \neq c(x)\right] &= \Pr_{x \sim D}\left[g_+(x) \neq c(x) \wedge x \in \mathrm{Rob}_{\mathcal{U}}(\hat{h})\right] + \Pr_{x \sim D}\left[g_+(x) \neq c(x) \wedge x \notin \mathrm{Rob}_{\mathcal{U}}(\hat{h})\right] \\
&= \Pr_{x \sim D}\left[\hat{h}(x) \neq c(x) \wedge x \in \mathrm{Rob}_{\mathcal{U}}(\hat{h})\right] + \Pr_{x \sim D}\left[g_+(x) \neq c(x) \wedge x \notin \mathrm{Rob}_{\mathcal{U}}(\hat{h})\right] \\
&\leq \Pr_{x \sim D}\left[\hat{h}(x) \neq c(x) \wedge x \in \mathrm{Rob}_{\mathcal{U}}(\hat{h})\right] + \Pr_{x \sim D}\left[x \notin \mathrm{Rob}_{\mathcal{U}}(\hat{h})\right] \leq \epsilon + \epsilon = 2\epsilon.
\end{aligned}
$$

Finally, observe that for any $x \in \mathrm{Rob}_{\mathcal{U}}(\hat{h})$ such that $\hat{h}(x) = +1$, by definition of $g_+$, it holds that $x \in \mathrm{Rob}_{\mathcal{U}^{-1}(\mathcal{U})}(g_+)$, thus

$$
\begin{aligned}
\Pr_{x \sim D}\left[x \in \mathrm{Rob}_{\mathcal{U}^{-1}(\mathcal{U})}(g_+)\right] &\geq \Pr_{x \sim D}\left[x \in \mathrm{Rob}_{\mathcal{U}}(\hat{h}) \wedge \hat{h}(x) = +1\right] \\
&= \Pr_{x \sim D}\left[x \in \mathrm{Rob}_{\mathcal{U}}(\hat{h})\right] \Pr_{x \sim D}\left[\hat{h}(x) = +1 \big| x \in \mathrm{Rob}_{\mathcal{U}}(\hat{h})\right] \\
&\geq (1 - \epsilon) \Pr_{x \sim D}\left[\hat{h}(x) = +1 \big| x \in \mathrm{Rob}_{\mathcal{U}}(\hat{h})\right].
\end{aligned}
$$

$\square$

*Proof of Theorem 4.* Let $\mathcal{U}$ be an arbitrary perturbation set, and $\mathbb{B}$ an $(\epsilon, \delta)$-robust learner for some *unknown* target concept $c : \mathcal{X} \to \mathcal{Y}$ with respect to $\mathcal{U}$. We will construct another learner $\tilde{\mathbb{B}}$ that $(\beta, 2\epsilon, 2\delta)$-barely-robustly-learns $c$ with respect to $\mathcal{U}^{-1}(\mathcal{U})$, with $\beta = (1 - \epsilon)/2$. Let $D$ be some unknown distribution over $\mathcal{X}$ that is robustly realizable: $\mathrm{Pr}_{x \sim D}\left[\exists z \in \mathcal{U}(x) : c(z) \neq c(x)\right] = 0$.

**Description of $\tilde{\mathbb{B}}$.** Sample $S \sim D_c^{m_{\mathbb{B}}(\epsilon, \delta)}$, and run learner $\mathbb{B}$ on $S$. Let $\hat{h} = \mathbb{B}(S)$ be the predictor returned by $\mathbb{B}$. Let $\tilde{m} \geq \frac{64}{9} \ln(1/\delta)$. For each $1 \leq i \leq \tilde{m}$, consider the following process: draw an example $(x, y) \sim D_c$. If $x \in \mathrm{Rob}_{\mathcal{U}}(\hat{h})$ terminate, otherwise repeat the process again. Let $\tilde{S} = \{(x_1, y_1), \ldots, (x_{\tilde{m}}, y_{\tilde{m}})\}$ be the sample resulting from this process. Calculate $M_+ = \frac{1}{|\tilde{S}|} \sum_{x \in \tilde{S}} \mathbb{1}[\hat{h}(x) = +1]$. If $M_+ \geq 1/2$, output $g_+$, otherwise, output $g_-$ (as defined in Lemma 5).

**Analysis.** With probability at least $1 - \delta$ over $S \sim D_c^m$, $\hat{h}$ has small robust risk: $\mathrm{R}_{\mathcal{U}}(\hat{h}; D_c) \leq \epsilon$. Lemma 5 implies then that for each $y \in \{\pm 1\}$, $g_y$ satisfies:

$$
\Pr_{x \sim D}\left[g_y(x) \neq c(x)\right] \leq 2\epsilon \text{ and } \Pr_{x \sim D}\left[x \in \mathrm{Rob}_{\mathcal{U}^{-1}(\mathcal{U})}(g_y)\right] \geq (1 - \epsilon) \Pr_{x \sim D}\left[\hat{h}(x) = y \big| x \in \mathrm{Rob}_{\mathcal{U}}(\hat{h})\right].
$$

It remains to show that with probability at least $1 - \delta$ over $\tilde{S} \sim D^{\tilde{m}}$, for $g_{\hat{y}}$ returned by $\tilde{\mathbb{B}}$:

$$
\Pr_{x \sim D}\left[\hat{h}(x) = \hat{y} \big| x \in \mathrm{Rob}_{\mathcal{U}}(\hat{h})\right] \geq \frac{1}{2}.
$$

Observe that by the rejection sampling mechanism of $\tilde{\mathbb{B}}$, $\tilde{S}$ is a sample from the region of distribution $D$ where $\hat{h}$ is robust. Furthermore, we know that

$$
\max\left\{\Pr_{x \sim D}\left[\hat{h}(x) = +1 \big| x \in \mathrm{Rob}_{\mathcal{U}}(\hat{h})\right], \Pr_{x \sim D}\left[\hat{h}(x) = -1 \big| x \in \mathrm{Rob}_{\mathcal{U}}(\hat{h})\right]\right\} \geq \frac{1}{2}.
$$

Without loss of generality, suppose that $p = \mathrm{Pr}_{x \sim D}\left[\hat{h}(x) = +1 \big| x \in \mathrm{Rob}_{\mathcal{U}}(\hat{h})\right] \geq 1/2$. Then, the failure event is that $\tilde{\mathbb{B}}$ outputs $g_-$, i.e. the event that $M_+ < \frac{1}{2}$. By a standard application of the Chernoff bound, we get that

$$
\Pr_{\tilde{S}}\left[M_+ < \frac{1}{2}\right] \leq e^{\frac{-\tilde{m} p \frac{1}{4}\left(\frac{1}{2} - \frac{1}{p}\right)^2}{2}} \leq e^{-\frac{9\tilde{m}}{64}} \leq \delta,
$$

where the last inequality follows from the choice of $\tilde{m}$ in the description of $\tilde{\mathbb{B}}$.

Finally, to conclude, observe that the sample complexity of learner $\tilde{\mathbb{B}}$ is equal to $m_{\mathbb{B}}(\epsilon, \delta)$ plus the number of samples drawn from $D$ to construct $\tilde{S}$. For each $1 \leq i \leq \tilde{m}$, let $X_i$ be the number of samples drawn from $D$ until a sample from the robust region $\text{Rob}_{\mathcal{U}}(\hat{h})$ was observed. Note that $X_i$ is a geometric random variable with mean at most $1/(1-\epsilon)$. By a standard concentration inequality for the sums of i.i.d. geometric random variables [4],

$$\Pr\left[\sum_{i=1}^{\tilde{m}} X_i > 2\frac{\tilde{m}}{1-\epsilon}\right] \leq e^{-\frac{\tilde{m}}{4}} \leq \delta,$$

where the last inequality follows from the choice of $\tilde{m}$ in the description of $\tilde{\mathbb{B}}$. Thus, with probability at least $1-\delta$, the total sample complexity is $m_{\mathbb{B}}(\epsilon, \delta) + \frac{2\tilde{m}}{1-\epsilon}$. This concludes that learner $\tilde{\mathbb{B}}$ $(\beta, 2\epsilon, 2\delta)$-barely-robustly-learns $c$ w.r.t. $\mathcal{U}^{-1}(\mathcal{U})$, where $\beta = (1-\epsilon)/2$. $\qquad\square$

## D  Proof of Theorem 6

**Definition 4** (Robust Shattering Dimension). A sequence $z_1, \ldots, z_k \in \mathcal{X}$ is said to be $\mathcal{U}$-*robustly shattered* by $\mathcal{C}$ if $\exists x_1^+, x_1^-, \ldots, x_k^+, x_k^- \in \mathcal{X}$ such that $\forall i \in [k], z_i \in \mathcal{U}(x_i^+) \cap \mathcal{U}(x_i^-)$ and $\forall y_1, \ldots, y_k \in \{\pm 1\} : \exists h \in \mathcal{C}$ such that $h(z') = y_i \forall z' \in \mathcal{U}(x_i^{y_i}), \forall 1 \leq i \leq k$. The $\mathcal{U}$-*robust shattering dimension* $\dim_{\mathcal{U}}(\mathcal{C})$ is defined as the largest $k$ for which there exist $k$ points $\mathcal{U}$-robustly shattered by $\mathcal{C}$.

The following lemma due to [15] states that finite robust shattering dimension $\dim_{\mathcal{U}}(\mathcal{C})$ is *necessary* for strongly robustly learning $\mathcal{C}$ with respect to $\mathcal{U}$.

**Lemma 10** ([15]). *For any class $\mathcal{C}$ and any perturbation set $\mathcal{U}$, $\mathcal{C}$ is $(\epsilon, \delta)$-robustly-learnable with respect to $\mathcal{U}$ only if $\dim_{\mathcal{U}}(\mathcal{C})$ is finite.*

We are now ready to proceed with the proof of Theorem 6.

*Proof of Theorem 6.* Pick three infinite unique sequences $(x_n^+)_{n \in \mathbb{N}}$, $(x_n^-)_{n \in \mathbb{N}}$, and $(z_n)_{n \in \mathbb{N}}$ from $\mathbb{R}^2$ such that for each $n \in \mathbb{N} : x_n^+ = (n, 1), x_n^- = (n, -1), z_n = (n, 0)$, and let $\mathcal{X} = \cup_{n \in \mathbb{N}} \{x_n^+, x_n^-, z_n\}$. We now describe the construction of the perturbation set $\mathcal{U}$. For each $n \in \mathbb{N}$, let $\mathcal{U}(x_n^+) = \{x_n^+, z_n\}, \mathcal{U}(x_n^-) = \{x_n^-, z_n\}$, and $\mathcal{U}(z_n) = \{z_n, x_n^+, x_n^-\}$.

We now describe the construction of the concept class $\mathcal{C}$. For each $\boldsymbol{y} \in \{\pm 1\}^{\mathbb{N}}$ define $h_{\boldsymbol{y}} : \mathcal{X} \to \mathcal{Y}$ to be:
$$\forall n \in \mathbb{N} : h_{\boldsymbol{y}}(z_n) = y_n \wedge h_{\boldsymbol{y}}(x_n^+) = +1 \wedge h_{\boldsymbol{y}}(x_n^-) = -1. \tag{12}$$

Let $\mathcal{C} = \left\{ h_{\boldsymbol{y}} : \boldsymbol{y} \in \{\pm 1\}^{\mathbb{N}} \right\}$. Observe that by construction of $\mathcal{U}$ and $\mathcal{C}$, $\mathcal{C}$ robustly shatters the sequence $(z_n)_{n \in \mathbb{N}}$ with respect to $\mathcal{U}$ (see Definition 4), and therefore, the robust shattering dimension of $\mathcal{C}$ with respect to $\mathcal{U}$, $\dim_{\mathcal{U}}(\mathcal{C})$, is infinite. Thus, Lemma 10 implies that $\mathcal{C}$ is not $(\epsilon, \delta)$-*strongly-robustly-learnable* with respect to $\mathcal{U}$.

We will now show that there is a simple learner $\mathbb{A}$ that $(\beta, \epsilon, \delta)$-*barely*-robustly-learns $\mathcal{C}$ with respect to $\mathcal{U}$, with robustness parameter $\beta = \frac{1}{2}$ and natural error $\epsilon = 0$. Specifically, $\mathbb{A}$ samples a bitstring $\tilde{\boldsymbol{y}} \in \{\pm 1\}^{\mathbb{N}}$ uniformly at random, and outputs the classifier $h_{\tilde{\boldsymbol{y}}}$. Learner $\mathbb{A}$ will not require any data as input.

We now proceed with analyzing the performance of learner $\mathbb{A}$. Let $h_{\boldsymbol{y}} \in \mathcal{C}$ be some unknown target concept and $D$ be some unknown distribution over $\mathcal{X}$ that is robustly realizable: $\Pr_{x \sim D}[\exists z \in \mathcal{U}(x) : h_{\boldsymbol{y}}(z) \neq y] = 0$. Since $D$ is robustly realizable, by construction of $\mathcal{U}$ and $\mathcal{C}$, this implies that
$$\forall n \in \mathbb{N} : D(z_n) = 0 \text{ and } D(x_n^{-y_n}) = 0. \tag{13}$$

This is because $\mathcal{U}(z_n) = \{z_n, x_n^+, x_n^-\}$ and Equation 12 implies that $h_{\boldsymbol{y}}$ is not robust on $z_n$ since $h_{\boldsymbol{y}}(x_n^+) \neq h_{\boldsymbol{y}}(x_n^-)$, also $\mathcal{U}(x_n^+) \cap \mathcal{U}(x_n^-) = \{z_n\}$ and since $h_{\boldsymbol{y}}(z_n) = y_n$ this implies that $h_{\boldsymbol{y}}$ is not robust on $x_n^{-y_n}$. Equation 13 and Equation 12 together imply that the random classifier $h_{\tilde{\boldsymbol{y}}} \in \mathcal{C}$ selected by learner $\mathbb{A}$ has *zero* error on *natural* examples: with probability 1 over $\tilde{\boldsymbol{y}}$, $\Pr_{x \sim D}[h_{\tilde{\boldsymbol{y}}}(x) \neq h_{\boldsymbol{y}}(x)] = 0$.

We now turn to analyzing the *robust* risk of learner $\mathbb{A}$,

$$
\begin{aligned}
\mathbb{E}_{\tilde{\boldsymbol{y}}} \left[ \mathbb{E}_{x \sim D} \left[ \mathbb{1}\left\{ \exists z \in \mathcal{U}(x) : h_{\tilde{\boldsymbol{y}}}(z) \neq h_{\boldsymbol{y}}(x) \right\} \right] \right] &= \mathbb{E}_{x \sim D} \left[ \mathbb{E}_{\tilde{\boldsymbol{y}}} \left[ \mathbb{1}\left\{ \exists z \in \mathcal{U}(x) : h_{\tilde{\boldsymbol{y}}}(z) \neq h_{\boldsymbol{y}}(x) \right\} \right] \right] \\
&= \sum_{n \in \mathbb{N}} D(x_n^{y_n}) \mathbb{E}_{\tilde{\boldsymbol{y}}} \left[ \mathbb{1}\left\{ \exists z \in \mathcal{U}(x_n^{y_n}) : h_{\tilde{\boldsymbol{y}}}(z) \neq h_{\boldsymbol{y}}(x_n^{y_n}) \right\} \right] \\
&= \sum_{n \in \mathbb{N}} D(x_n^{y_n}) \mathbb{E}_{\tilde{\boldsymbol{y}}} \left[ \mathbb{1}\left\{ h_{\tilde{\boldsymbol{y}}}(z_n) \neq h_{\boldsymbol{y}}(x_n^{y_n}) \right\} \right] \\
&= \sum_{n \in \mathbb{N}} D(x_n^{y_n}) \mathbb{E}_{\tilde{\boldsymbol{y}}} \left[ \mathbb{1}\left\{ \tilde{y}_n \neq y_n \right\} \right] = \sum_{n \in \mathbb{N}} D(x_n^{y_n}) \frac{1}{2} = \frac{1}{2}.
\end{aligned}
$$

This implies that in expectation over randomness of learner $\mathbb{A}$, it will be robust on half the mass of distribution $D$: $\mathbb{E}_{\tilde{\boldsymbol{y}}} \mathbb{E}_{x \sim D} \mathbb{1}[x \in \mathrm{Rob}_{\mathcal{U}}(h_{\tilde{\boldsymbol{y}}})] = \frac{1}{2}$. $\qquad\square$

## E  Robustness at Different Levels of Granularity

For concreteness, throughout the rest of this section, we consider robustness with respect to metric balls $\mathrm{B}_\gamma(x) = \{z \in \mathcal{X} : \rho(x, z) \leq \gamma\}$ where $\rho$ is some metric on $\mathcal{X}$ (e.g., $\ell_\infty$ metric), and $\gamma > 0$ is the perturbation radius. Achieving small robust risk with respect to a *fixed* perturbation set $\mathrm{B}_\gamma$ is the common goal studied in adversarially robust learning. What we studied so far in this work is learning a predictor $\hat{h}$ robust to $\mathcal{U} = \mathrm{B}_\gamma$ perturbations as measured by the robust risk: $\mathrm{Pr}_{(x,y) \sim D} \left[ \exists z \in \mathrm{B}_\gamma(x) : \hat{h}(z) \neq y \right]$, when given access to a learner $\mathbb{A}$ barely robust with respect to $\mathcal{U}^{-1}(\mathcal{U}) = \mathrm{B}_{2\gamma}$.

Our original approach to boosting robustness naturally leads us to an alternate interesting idea: learning a cascade of robust predictors with different levels of granularity. This might be desirable in situations where it is difficult to robustly learn a distribution $D_c$ over $\mathcal{X} \times \mathcal{Y}$ with robustness granularity $\gamma$ everywhere, and thus, we settle for a weaker goal which is first learning a robust predictor $h_1$ with granularity $\gamma$ on say $\beta$ mass of $D$, and then recursing on the conditional distribution of $D$ where $h_1$ is not $\gamma$-robust and learning a robust predictor $h_2$ with granularity $\gamma/2$, and so on. That is, we are adaptively learning a sequence of predictors $h_1, \ldots, h_T$ where each predictor $h_t$ is robust with granularity $\frac{\gamma}{2^t}$. Furthermore, if we are guaranteed that in each round we make progress on some $\beta$ mass then it follows that

$$
\mathrm{Pr}\left[ \cup_{t=1}^{T} \mathrm{Rob}_{\gamma/2^{t-1}}(h_t) \right] = 1 - \mathrm{Pr}\left[ \cap_{t=1}^{T} \overline{\mathrm{Rob}}_{\gamma/2^{t-1}}(h_t) \right] \geq 1 - (1 - \beta)^T,
$$

and the cascade predictor $\mathrm{CAS}(h_{1:T})$ has the following robust risk guarantee

$$
\sum_{t=1}^{T} \mathrm{Pr}\left[ \bar{R}_{1:t-1} \wedge \left( \exists z \in \mathrm{B}_{\gamma/2^t}(x) : \forall_{t' < t} G_{h_{t'}}(z) = \perp \wedge G_{h_t}(z) = 1 - c(x) \right) \right] + \mathrm{Pr}\left[ \bar{R}_{1:T} \right] \leq \frac{\epsilon}{\beta} + (1 - \beta)^T.
$$

In words, the cascade predictor $\mathrm{CAS}(h_{1:T})$ offers robustness at different granularities. That is, for $x \sim D$ such that $x \in R_1$, $\mathrm{CAS}(h_{1:T})$ is guaranteed to be robust on $x$ with granularity $\gamma$, and for $x \sim D$ such that $x \in \bar{R}_1 \cap R_2$, $\mathrm{CAS}(h_{1:T})$ is guaranteed to be robust on $x$ with granularity $\gamma/2$, and so on.

**Applications.**  We give a few examples where this can be useful. Consider using SVMs as barely robust learners. SVMs are known to be margin maximizing learning algorithms, which is equivalent to learning linear predictors robust to $\ell_2$ perturbations. In our context, by combining SVMs with our boosting algorithm, we can learn a cascade of linear predictors each with a maximal margin on the conditional distribution.

## F Python Code

```python
# MNIST odd vs. even
from torchvision import datasets, transforms
from sklearn.preprocessing import StandardScaler
from sklearn.pipeline import make_pipeline
from sklearn.svm import LinearSVC

def flatten(X):
    return X.reshape(X.shape[0], -1)

train_set = datasets.MNIST('./', train=True, download=True, transform
    = transforms.Compose([transforms.ToTensor()]))
test_set = datasets.MNIST('./', train=False, download=True, transform
    = transforms.Compose([transforms.ToTensor()]))

X_train = train_set.train_data.numpy()
y_train = train_set.train_labels.numpy()
X_test = test_set.test_data.numpy()
y_test = test_set.test_labels.numpy()
# Convert to binary labels
y_train_binary = 2*np.array([y % 2 for y in y_train])-1
y_test_binary = 2*np.array([y % 2 for y in y_test])-1

#flatten data
f_X_tr = flatten(X_train) / 255.0
f_X_te = flatten(X_test) / 255.0

# sample size, and robustness radius
m, gamma, T = 10000, 1.0, 2

# fit a linear classifier
print('===================================================')
print('Baseline Linear Classifier:')
base_clf = LinearSVC(C=10e-7, fit_intercept=False, loss='hinge', tol=1
    e-5)
base_clf.fit(f_X_tr, y_train_binary)
print("0-1 Accuracy - Testing:", 100*base_clf.score(f_X_te,
    y_test_binary))

# robust accuracy
y_margin = y_test_binary*(base_clf.decision_function(f_X_te) / np.
    linalg.norm(base_clf.coef_))
robust_accuracy = len(y_margin[y_margin >= gamma]) / len(y_margin)
print('Robust Accuracy - Test:', 100*robust_accuracy)

y_margin = y_train_binary*(base_clf.decision_function(f_X_tr) / np.
    linalg.norm(base_clf.coef_))
robust_accuracy = len(y_margin[y_margin >= gamma]) / len(y_margin)
print('===================================================')

# boosting rounds
print('\nBoosting Robustness')
clf = []
for t in range(1,T+1):
    print("round ", t)
    # SVM
    lin_clf = LinearSVC(C=10e-7, fit_intercept=False, loss='hinge',
        tol=1e-5)
    lin_clf.fit(f_X_tr, y_train_binary)
    clf.append(lin_clf)

    if t < T:
        # Compute margin on fresh sample
```

```python
        margin = lin_clf.decision_function(f_X_tr) / np.linalg.norm(
            lin_clf.coef_)
        amargin = abs(margin)
        print("Fraction of Training Data with Robustness at least 2
            gamma: ", 100*float(len(amargin[amargin >= 2*gamma])/len(
            amargin)))
        print(len(amargin[amargin < 2*gamma]), " samples with small
            margin")

        # Update / Filter training data
        f_X_tr = f_X_tr[amargin < 2*gamma]
        y_train_binary = y_train_binary[amargin < 2*gamma]

# Evaluation
print('\nEvaluation - Test')
print('number of test samples', len(y_test_binary))
adv_mistake = 0
total = len(y_test_binary)
for t in range(0,T):
    print('round ', t)
    amargin = abs(clf[t].decision_function(f_X_te) / np.linalg.norm(
        clf[t].coef_))
    y_margin = y_test_binary*(clf[t].decision_function(f_X_te) / np.
        linalg.norm(clf[t].coef_))
    adv_mistake += len(y_test_binary[y_margin < -1*gamma])
    print('adversarial mistakes', len(y_test_binary[y_margin < -1*
        gamma]))

    f_X_te = f_X_te[(y_margin < 2*gamma) & (y_margin >= -1*gamma)]
    y_test_binary = y_test_binary[(y_margin < 2*gamma) & (y_margin >=
        -1*gamma)]
    print('abstained on data', len(y_test_binary), len(f_X_te))

    # last round
    if t == T-1 and len(f_X_te)>0:
        y_margin = y_test_binary*(clf[t].decision_function(f_X_te) /
            np.linalg.norm(clf[t].coef_))
        print('classifications in final round', len(y_margin))
        ee = len(y_test_binary[(y_margin < gamma)])
        print('extra mistakes', ee)
        adv_mistake += ee

print('total number of adv. mistakes', adv_mistake)
print('================================================')
print('Cascade Robust Accuracy:', 100*float((total - adv_mistake)/
    total) )
print('================================================')
```