# OpenReview forum: "Boosting Barely Robust Learners: A New Perspective on Adversarial Robustness"
_NeurIPS.cc/2022/Conference — NeurIPS 2022 Accept_

### Official Review · Reviewer_DokF · 2022-06-21

**Rating:** 8
**Confidence:** 5
**Soundness:** 4 excellent
**Presentation:** 3 good
**Contribution:** 4 excellent

**Summary:**

This paper mainly shows that strongly robust learnable w.r.t. $\mathcal{U}$ is equivalent to barely robust learable w.r.t. $\mathcal{U}^{-1}(\mathcal{U})$, bu giving an algorithm $\beta$-RoBoost to boost robustness of barely robust learners and an algorithm to return a barely robust classifier through calling the strongly robust learner.

This paper also shows that barely robust learable w.r.t. $\mathcal{U}$ is not sufficient for strongly robust learnable w.r.t. $\mathcal{U}$ by a counterexample. What's more, the paper also gives algorithms to boost robustness with unlabeled data.

**Questions:**

In the proof of Theorem 6, the paper uses a randomized learner which do not need any data as input, and proves that it is $(\frac{1}{2},0,\delta)$-barely robust learnable. And the paper proves that the expected value of the **robust risk** w.r.t. the random choice of $\tilde{y}$ is $\frac{1}{2}$, but not to be $\frac{1}{2}$ with probability at least $1 - \delta$ over the choice of samples as in the definition of the barely robust learner. Although we can use concentration inequality to get a confidence, but it is over the random choice of $\tilde{y}$, so there seems to be a mismatch in the definition of the barely robust learner and the proof of Theorem 6.

In Corollary 2, the paper claims it is a stronger boosting guarantee, I wonder in what sense is it stronger than Thoerem 1? Is it falls on better sample complexity, or falls on weaker requirement on the barely robust learner, or others else?

**Limitations:**

The authors does.

**Strengths And Weaknesses:**

Strengths:
- This paper shows a sufficient and necessary condition for strongly robust learning and gives a novelty method to boost robustness of barely robust learners. The theoretical results are quite solid.
- The results of this paper in significant since it provides another way to study the robustness.
- The paper gives some high-level strategies to help understanding the spirit in the algorithms and proofs.

Weaknesses:
- As discussed in the discussion part, verifying whether a classifier is robust on the perturbation set $\mathcal{U}^{-1}(\mathcal{U})$ is a key issue in the computational efficiency of the proposed algorithm, but **to the best of my knowledge, there is now no theoretiacl gurantee for precisely verifying robustness for a general perturbation set and a general hypothesis class**. As stated in [1], the popular method random smooth might fail in verifying the $\ell_\infty$ robustness for high-dimensional images. So the realizability of the proposed algorithm highly depends on the development of robust verification.

[1] Blum, Avrim, et al. "Random Smoothing Might be Unable to Certify $\ell_\infty$ Robustness for High-Dimensional Images." J. Mach. Learn. Res. 21 (2020): 211-1.

---

> ### Author Response · Authors · 2022-08-02
> **Response to Reviewer DokF**
>
> We thank you for your detailed review and valuable feedback.
>
> > As discussed in the discussion part, verifying whether a classifier is robust on the perturbation set $U^{-1}(U)$ is a key issue in the computational efficiency of the proposed algorithm, but to the best of my knowledge, there is now no theoretical guarantee for precisely verifying robustness for a general perturbation set and a general hypothesis class. As stated in [1], the popular method random smooth might fail in verifying the  robustness for high-dimensional images. So the realizability of the proposed algorithm highly depends on the development of robust verification.
>
> Our view on this is that, in general, it's hard to imagine a world where efficient robust learning is possible without efficient robust verification. In other words, efficient robust verification seems to be a "necessary" condition for efficient robust learning, not just for our proposed algorithm, but for any algorithm that is devised for robust learning. Without efficient robust verification, we can't even compute the robust loss, so how can we even learn.
>
> > In the proof of Theorem 6, the paper uses a randomized learner which do not need any data as input, and proves that it is ($\frac{1}{2}, 0, \delta$)-barely robust learnable. And the paper proves that the expected value of the robust risk w.r.t. the random choice of is $\tilde{y}$, but not to be with probability at least $1-\delta$ over the choice of samples as in the definition of the barely robust learner. Although we can use concentration inequality to get a confidence, but it is over the random choice of $\tilde{y}$, so there seems to be a mismatch in the definition of the barely robust learner and the proof of Theorem 6.
>
> Thank you for noticing this mismatch. Indeed, the current Definition 1 doesn’t take into account randomized learning algorithms. We will update it to allow randomized learning algorithms. Specifically, in line 41 in Definition 1, we just need to change "with probability at least 1 − δ over S" to "with probability at least 1 − δ over S and any internal randomness in the algorithm”, to allow us to use a randomized construction in the proof of Theorem 6. Then, as you mentioned, using a concentration inequality, we can get a high probability guarantee that the robust risk of the output predictor of the randomized learner in Theorem 6 is at most ½. We will update the paper to reflect this change. We also emphasize that this change doesn't affect $\beta$-RoBoost in the reverse direction (Theorem 1) since failure is failure, i.e., we already account for such failure (see lines 244-256).
>
> > In Corollary 2, the paper claims it is a stronger boosting guarantee, I wonder in what sense is it stronger than Thoerem 1? Is it falls on better sample complexity, or falls on weaker requirement on the barely robust learner, or others else?
>
> It is “stronger” than Theorem 1, because it requires a “weaker” barely-robust-learner with parameters ($\beta, \frac{\beta}{6}$) instead of ($\beta$, $\frac{\beta\epsilon}{2}$). Note that this avoids dependence on $\epsilon$ (the target robust risk that we want to achieve). We also ask open questions in lines 311-320, on whether it’s possible to improve upon this further.

---

> > ### Comment · Reviewer_DokF · 2022-08-03
> > **feedback to the response**
> >
> > Thank you for your valuable feedback, I think you have solved my questions, I think the paper is very good and I am willing to increase the score.

---

### Official Review · Reviewer_VncC · 2022-07-08

**Rating:** 7
**Confidence:** 4
**Soundness:** 4 excellent
**Presentation:** 4 excellent
**Contribution:** 3 good

**Summary:**

This paper studies how to achieve adversarially robust learning with weak robust learners. The investigation follows a quite natural agenda and gives quite complete answers. The contributions are three folds.

First, at the conceptual level, the paper propose the key definition of barely robust learning, and point out the importance to look at a larger perturbation set $U^{-1}(U)$.

Second, the paper gives an oracle efficient reduction from adversarially robust learning to barely robust learning. This shows that barely robust learning with respect to $U^{-1}(U)$ is sufficient for adversarial robust learning. This is by Theorem 1 and Algorithm 1.

Third, the paper shows that adversarial robust learning implies barely robust learning with respect to $U^{-1}(U)$ (Theorem 4), but it can not be achieved if one only has barely robust learning with respect to $U$ (Theorem 6). This shows that barely robust learning with respect to $U^{-1}(U)$ is necessary for adversarially robust learning.

**Questions:**

I am confident that I understand the theme and conclusions of the paper. But I only go through the algorithm and proof part, so it's hard for me to evaluate the novelty of the algorithm and the theoretical analysis. I think the concept of barely robustness is quite essential, but is the analysis somewhat similar to existing routines after one discover this concept?

**Limitations:**

The proposed algorithm in the paper is more useful when the robust parameter $\beta$ is small, since existing results could not provide theoretical guarantees in this regime. However, experiments may be needed if the authors want to advertise their algorithm to be better than existing methods that are directly based on canonical results and already work for $\beta>1/2$.

I think the paper should put more space and content to discuss when the perturbation set $U^{-1}(U)$ is much larger than $U$, and when they are similar. The current paper only uses the $\ell_p$ ball examples, where requiring robustness with respect to $U^{-1}(U)$ is very reasonable (it's just a ball with twice radius). However, as looking at this larger perturbation set is one of the major novelty of the paper, the author should carefully explain for what $U$ this requirement is strong (which means boosting robustness is not easy, according to the theorems in the paper), and for what $U$ this requirement is weak (which means boosting robustness is easy).

**Strengths And Weaknesses:**

I like the flow of writing in this paper, especially the introduction section. Although this is a purely theoretical paper, the pace makes its proposed questions and main contributions easy to understand. In particular, when I read the introduction, the first question coming to mind is, isn't boosting easier if the robust parameter $\beta$ is larger than 1/2? This is answered immediately in the paragraph "Landscape of boosting robustness." The second question coming to mind is, is the assumption of robust realizability more restrictive than the traditional concept of realizability. And the authors give examples to answer this question in the related work part. I also like the  discussion section (Section 5), which answers my concerns about the agnostic setting and computation efficiency. This part also propose several open questions.

---

> ### Author Response · Authors · 2022-08-02
> **Response to Reviewer VncC**
>
> We thank you for your detailed review and valuable feedback.
>
> > I am confident that I understand the theme and conclusions of the paper. But I only go through the algorithm and proof part, so it's hard for me to evaluate the novelty of the algorithm and the theoretical analysis. I think the concept of barely robustness is quite essential, but is the analysis somewhat similar to existing routines after one discover this concept?
>
> We would like to emphasize that besides the conceptual contribution of the definition of barely-robust-learners wrt $U^{-1}(U)$ (Definition 1), the algorithmic contributions are novel such as doing rejection sampling and focusing on the region of non-robustness. Additionally, it is not immediately obvious how to combine the predictors $h_1,\dots, h_T$ after $T$ rounds of boosting robustness, and we propose one specific mechanism for selective classification (line 141 and Equation 4) that essentially guarantees that each predictor makes predictions only in the region where it is robust. Beyond this, the analysis of the robust risk (lines 234-243) is not trivial, and requires careful thought about what the adversary can do at test-time. For example, earlier we had an analysis that required a barely-robust-learner with parameters ($\beta$, $\frac{\epsilon}{2T}$), where there’s dependence on the number of rounds, but later we have improved the analysis such that only ($\beta$, $\frac{\beta\epsilon}{2}$) is needed.
>
> > The proposed algorithm in the paper is more useful when the robust parameter $\beta$ is small, since existing results could not provide theoretical guarantees in this regime. However, experiments may be needed if the authors want to advertise their algorithm to be better than existing methods that are directly based on canonical results and already work for $\beta > \frac{1}{2}$.
>
> We believe that our boosting algorithm complements previous classical boosting algorithms in a beautiful way, rather than compete with them. One instantiation of this is Corollary II (line 162), which shows how to arrive at a stronger boosting guarantee by combining $\beta$-RoBoost with classical boosting algorithms. Nonetheless, we will provide simple experiments in a separate comment illustrating the utility of our algorithm in the regime where $\beta < \frac{1}{2}$.
>
> > I think the paper should put more space and content to discuss when the perturbation set $U^{-1}(U)$ is much larger than $U$, and when they are similar. The current paper only uses the $\ell_p$ ball examples, where requiring robustness with respect to $U^{-1}(U)$ is very reasonable (it's just a ball with twice radius). However, as looking at this larger perturbation set is one of the major novelty of the paper, the author should carefully explain for what $U$ this requirement is strong (which means boosting robustness is not easy, according to the theorems in the paper), and for what $U$ this requirement is weak (which means boosting robustness is easy).
>
> Thank you for suggesting comparing $U^{-1}(U)$ and $U$. As you mentioned, in the paper we give the simple example concerning general metric balls, if $U$ is a metric-ball of radius $\gamma$, then $U^{-1}(U)$ is a metric-ball of radius $2\gamma$. More generally, we think of robustness w.r.t. $U^{-1}(U)$ as being robust to “twice” the adversary’s power (which is represented by $U$). For example, for a general vector space $X$, if $U(x) = x + B$, where $B$ is some arbitrary symmetric set (i.e. $v \in B$ iff $-v\in B$) then $U^{-1}(U)(x) = x + B + B$.
>
> We do not know yet of examples of $U$ where $U^{-1}(U)$ is much “larger” than $U$, but this is an interesting question. At the same time, we don’t think that there is an easy vs. hard distinction here, since our results show that strong-robust-learning w.r.t $U$ implies barely-robust-learning w.r.t. $U^{-1}(U)$. In other words, boosting robustness is “easy” whenever strong-robust-learning w.r.t. $U$ is “easy”.

---

> > ### Comment · Reviewer_VncC · 2022-08-08
> > **Feedback to response**
> >
> > Thank you for your response. I think the paper is qualified for acceptance and maintain my score. The experiments will be helpful for broad audience, so please at least add them into the appendix if the paper gets acceptance.

---

### Official Review · Reviewer_TbrQ · 2022-07-11

**Rating:** 4
**Confidence:** 3
**Soundness:** 3 good
**Presentation:** 3 good
**Contribution:** 2 fair

**Summary:**

The paper studies a new direction in adversarial training, on the distinctions between barely robust leaners and strongly robust learners. The paper provides theoretical results on whether we can make a barely robust classifier into a strongly robust classifier. The paper also provided two algorithms to guarantee a strongly robust learner. The authors also prove that for any distribution, there would exist a barely robust classifier and that classifier is not a strongly robust classifier.

**Questions:**

1. You mentioned that  $U^{-1}(U)$ is a ball with $2\gamma$ radius and $U$ is a ball with $\gamma$ radius. This statement and Colloray I sound contradicitng with you claim that "weaker relaxations of barely robust learning do not imply strongly robust learning" in line 33.
2. Is it possible to implement black-box oracle calls to A on real-world datasets?
3.

**Limitations:**

Although the theoretical results are novel and interesting, the paper does not provide numerical experiments to support its findings. This makes me question the possibility of using the definition of the strongly robust learner or implementing the two algorithms in real-world applications. It would be more convincing if the authors can provide some numerical experiments even on small datasets, like MNIST.

**Strengths And Weaknesses:**

Strengths:
1. The definition of a barely robust learner is solid and new.
2. The authors provide many theorems to explain the relationship between a barely robust classifier and a strongly robust classifier, including their equivalence under some conditions.
3. There are two algorithms that guarantee getting a strongly robust classifier.

Weaknesses:
1. It is still unclear if the barely robust classifier for $U^{-1}(U)$ and strongly robust classifier $U$ are equivalent, is it that important to train a strongly robust classifier? It is unclear why we would care about the definition of a strongly robust learner.
2. There are no numerical experiments to support the applications or benefits of a strongly robust classifier.
3. There are no comparisons between existing literature in either theoretical or numerical results, so we don't understand what is the relationship between a strongly robust classifier with other related literature.

---

> ### Author Response · Authors · 2022-08-02
> **Response to Reviewer TbrQ**
>
> We thank you for your review and feedback. We address below the concerns raised in your review, and clarify our results and contributions further.
>
> Our contributions are:
>
> Theorem 6: barely-robust-learning w.r.t. $U$ does NOT imply strong-robust-learning w.r.t. $U$. This is a specific construction that highlights that in general we can’t boost robustness (from $U$ to $U$) when $\beta\leq\frac{1}{2}$.
>
> Theorem 1: barely-robust-learning wrt $U^{-1}(U)$ implies strong-robust-learning wrt U. This is achieved using our $\beta$-RoBoost algorithm.
>
> Theorem 4: strong-robust-learning wrt $U$ implies barely-robust-learning wrt $U^{-1}(U)$.
>
> > It is still unclear if the barely robust classifier for $U^{-1}(U)$ and strongly robust classifier $U$ are equivalent,
>
> Theorem 1 and 4 together imply that barely-robust-learning wrt $U^{-1}(U)$ is equivalent to strong-robust-learning wrt $U$ (see Corollary I), so we don’t understand this comment but we would be happy to discuss this further.
>
> > is it that important to train a strongly robust classifier? It is unclear why we would care about the definition of a strongly robust learner.
>
> We want to clarify that our ultimate goal is to learn predictors that are robust to adversarial examples, where the robust risk is formally defined in Eqn 1 (line 13). strong-robust-learning (Definition 2) formalizes this objective in the spirit of PAC learning. We have summarized in Section 1.2 (lines 122-130) previous work that studied Definition 2, and described their results and discussed the relationship of that line of work to our work.
>
> > There are no comparisons between existing literature in either theoretical or numerical results, so we don't understand what is the relationship between a strongly robust classifier with other related literature.
>
> We actually DO compare our results, as noted by Reviewer Lbb7, with previous theoretical results in the literature in Section 1.2 (see lines 117-130). Furthermore, we also show how combining our main algorithmic result ($\beta$-RoBoost) with classical boosting algorithms yields even stronger guarantees (see lines 78-96 and Corollary II). Such a result would not have been possible if we ignored earlier results in the literature.
>
> > You mentioned that $U^{-1}(U)$ is a ball with $2\gamma$ radius and $U$ is a ball with $\gamma$ radius. This statement and Colloray I sound contradicitng with you claim that "weaker relaxations of barely robust learning do not imply strongly robust learning" in line 33.
>
> There is NO contradiction here. Perhaps the wording in line 33 is a bit confusing, but what we mean is: barely-robust-learning wrt $U^{-1}(U)$ is necessary for strong-robust-learning wrt $U$, and we can NOT relax it to barely-robust-learning wrt $U$ (which is a weaker requirement) when $\beta \leq \frac{1}{2}$.
>
> > There are no numerical experiments to support the applications or benefits of a strongly robust classifier.
>
> > Is it possible to implement black-box oracle calls to A on real-world datasets?
>
> > It would be more convincing if the authors can provide some numerical experiments even on small datasets, like MNIST.
>
> We will include in a separate comment some simple experiments that illustrate the utility of our theory for boosting robustness on MNIST.
>
> Finally, a minor comment on terminology: in our paper, barely-robust and strongly-robust are definitions concerning learning algorithms, and NOT classifiers.

---

### Official Review · Reviewer_YxMp · 2022-07-11

**Rating:** 3
**Confidence:** 2
**Soundness:** 2 fair
**Presentation:** 1 poor
**Contribution:** 2 fair

**Summary:**

This paper presents theoretical analysis over barely robust learner and strong robust learner.

**Questions:**

Could we add an empirical analysis to show that the proposed methods are effective?

**Limitations:**

I suggest the authors to discuss how the users can adapt the proposed algorithms in real applications.

**Strengths And Weaknesses:**

Strong Points
----
1. Adversarial robustness analysis is an important problem in machine learning community.

Weak Points
----
1. I spent more than 8 hours to read the paper. However, at least for me, it is very hard to follow the analysis. For example, U^{-1} is heavily used in Section 1.1 while its description locates in Section 2.
2. I didn't check the correctness of the theorems and proofs. I wonder if it is possible to add an empirical analysis to show that the proposed methods are effective? The termination rounds could be very large for real applications.

---

> ### Author Response · Authors · 2022-08-02
> **Response to Reviewer YxMp**
>
> We thank you for your time and reading our paper. We have strived to make the presentation clear, and we were happy that the other reviewers found it to be such, but we will of course work on making it clearer still.
>
> > For example, U^{-1} is heavily used in Section 1.1 while its description locates in Section 2.
>
> We chose to describe $U^{-1}(U)$ in lines 43-45 in Section 1.1 in words and with examples, and defer the full definition to Section 2. We didn’t want to bombard the reader with too many definitions before stating our main results.
>
> > I suggest the authors to discuss how the users can adapt the proposed algorithms in real applications.
>
> We discuss at a very high-level in lines 295-310 how our proposed boosting algorithm can be used in real-world applications. In this discussion, we pose the question of whether current adversarial learning methods satisfy the barely robust learning condition on real world datasets. This is an interesting empirical direction to explore, but beyond the scope of our theoretical work.
>
> > Could we add an empirical analysis to show that the proposed methods are effective?
>
> Yes, we will include in a separate comment some simple experiments that illustrate the utility of our theory for boosting robustness.

---

### Official Review · Reviewer_Lbb7 · 2022-07-12

**Rating:** 7
**Confidence:** 3
**Soundness:** 4 excellent
**Presentation:** 4 excellent
**Contribution:** 4 excellent

**Summary:**

The authors introduce a notion of \emph{barely robust} learning. Their main results demonstrate the an equivalence with strongly robust learning by showing (Theorem 4) that if a concept is strongly robustly learnable, it is barely robustly learnable as well, and then by proposing an oracle-efficient algorithm (Theorem 1) that calls a barely robust learner in order to produce a strongly robust one. Their formal equivalence statement (Corollary I) shows that being strongly robustly learnable with respect to an adversarial perturbation set $\mathcal{U}$ is equivalent to being barely robustly learnable with respect to a larger perturbation set $\mathcal{U}^{-1}(\mathcal{U})$, and they further demonstrate (Theorem 6) that barely robust learnability w.r.t. $\mathcal{U}$ does not in general imply strong robust learnability.



**Questions:**

See comments under Significance above. Any enlightenment on either of these aspects would greatly improve the paper.

**Limitations:**

The authors have not discussed potential negative societal impact of their work, although it is hard to see what one might have to say here that is not totally banal.

**Strengths And Weaknesses:**

Originality:
Highly original; the authors have proposed the first algorithm that boosts the robustness of what they call barely robust learners, and provide a comprehensive characterization of their newly proposed notion as it relates to previously studied notions.

Quality:
Good. The theory is elegantly laid out, and comprehensively addresses the salient questions. The discussion provides a satisfying contextualization of the results.

Clarity:
Generally quite good, although perhaps it would read more straightforwardly if relevant definitions were placed before they are required in theorem statements. (I believe there is a typo in line 151: a missing subscript t in the def. of $G_{h_t}(z)$.)

Significance:
The open question of whether one can boost the error of a barely robust learner without sacrificing robustness remains, it would seem, an impediment to greater significance. In the same vein, more insight into the possibility of currently known adversarial learning methods being barely robust would also be welcome.

---

> ### Author Response · Authors · 2022-08-02
> **Response to Reviewer Lbb7**
>
> We thank you for your detailed review and valuable feedback. We will work on improving the presentation further, and fix the typos.
>
> Regarding boosting the error of a barely-robust-learner without sacrificing robustness: Corollary II (line 162) says that we can boost a ($\beta$, $\frac{\beta}{6}$) barely-robust-learner to an $\epsilon$–strong-robust-learner. This is achieved by combining our algorithm $\beta$-RoBoost with classical boosting algorithms. This is an improvement over the result in Theorem 1 in the sense that our starting point of a barely-robust-learner is ($\beta$, $\frac{\beta}{6}$) instead of ($\beta$, $\frac{\beta\epsilon}{2}$) (i.e. we avoid dependence on target robust error $\epsilon$ here). It is still an open question whether we can remove dependence on $\beta$ in the error parameter (e.g. start from ($\beta$, $1/3$)) and we discuss this in lines (311-320). We think that one way to approach this is to try and boost ($\beta$,$1/3$) to a ⅓-strong-robust-learner, because then using Lemma 8 and Theorem 4, imply that we can get to a ($\frac{1-\epsilon}{2}, \epsilon$) barely-robust-learner. Another way would be to try and directly boost ($\beta, 1/3$) to ($\beta$, $\epsilon$). We think both directions can lead to interesting algorithmic consequences.
>
> Regarding adversarial learning methods satisfying the barely-robust-learning condition, we think that this can be distribution-dependent, i.e. perhaps it's true for some real world datasets but not others.

---

### Author Response · Authors · 2022-08-02
**Simple Experiments**

Based on requests from Reviewers YxMP and TbrQ, we have conducted simple experiments that illustrate the utility of our theoretical contribution for boosting robustness. These experiments demonstrate that our algorithm, $\beta$-RoBoost, can boost and improve the robustness of black-box learning algorithms. We describe the setup and the results below.

**Datasets.** A synthetic binary classification dataset (make_moons from scikit learn), and MNIST (rescaled by dividing by 255, and converted to binary classification of odd vs. even).

**Perturbation set $U$.** We consider $\ell_2$ perturbations of some radius $\gamma$. In the respective datasets, we computed the minimum distance between examples from different classes and chose a radius $\gamma$ that’s smaller than this minimum distance.

**Black-box Learning Algorithm.** We use an off-the-shelve Linear SVM solver (from scikit learn) as a barely-robust-learner. Using linear predictors and $\ell_2$ perturbations simplifies the computation of the robust loss since it exactly corresponds to computing the margin loss. Formally, for any linear predictor $w$ and example $(x,y)$,

$\sup_{\lVert \delta \rVert_2 \leq \gamma} \mathbf{1}[{\rm sign}(\langle w, x +\delta \rangle) \neq y] = \mathbf{1}[y \langle \frac{w}{\lVert w \rVert_2}, x \rangle  \leq \gamma].$

We ran our boosting algorithm, $\beta$-RoBoost, and compared it against the baseline of a single SVM call. In our boosting algorithm, we run for as many rounds as possible until there are no more examples left in the training set to run LinearSVM on.

**Pseudo-code.**

Baseline.
> Call LinearSVM on (X_train, y_train), and let $h$ be the returned linear predictor.

> Evaluate the robust accuracy of h on (X_test, y_test).

Beta-RoBoost.
> For $t=1$ to $T$:

> Run LinearSVM on (X_train, y_train), and let $h_t$ be the returned linear predictor.

> Filter (X_train, y_train) to only include examples where $h_t$ is not robust on radius $2\gamma$, i.e. only include examples with margin less than $2\gamma$.

> Evaluate the robust accuracy of the cascade predictor: $h_1, …, h_T$.

**Results:**

*Make Moons* with perturbation radius $\gamma=0.1$:

Baseline Linear SVM: achieves robust accuracy of 84.78%.

Our approach (boosting SVM for 2 rounds): achieves robust accuracy of **89.86%**.

*MNIST* with perturbation radius $\gamma=0.5$.

Baseline Linear SVM: achieves robust accuracy of 73.9%

Our algorithm (boosting SVM for 5 rounds): achieves robust accuracy of **80.05%**.


*MNIST* with perturbation radius $\gamma=1.0$.

Baseline Linear SVM: achieves robust accuracy of 48.1%.

Our approach (boosting SVM for 2 rounds): achieves robust accuracy of **70.12%**.

We observe that $\beta$-RoBoost improves the robustness of LinearSVM. Notice that even in the regime where the baseline LinearSVM archives robust-accuracy < 50% (MNIST with perturbation radius 1.0), $\beta$-RoBoost can actually improve the robust-accuracy.

---

> ### Author Response · Authors · 2022-08-02
> **Code**
>
> We include our code below to reproduce our MNIST experiments with perturbation radius $\gamma = 1.0$.
>
> ```
> # MNIST odd vs. even
> from torchvision import datasets, transforms
> from sklearn.preprocessing import StandardScaler
> from sklearn.pipeline import make_pipeline
> from sklearn.svm import LinearSVC
>
> def flatten(X):
>     return X.reshape(X.shape[0], -1)
>
> train_set = datasets.MNIST('./', train=True, download=True, transform = transforms.Compose([transforms.ToTensor()]))
> test_set = datasets.MNIST('./', train=False, download=True, transform = transforms.Compose([transforms.ToTensor()]))
>
> X_train = train_set.train_data.numpy()
> y_train = train_set.train_labels.numpy()
> X_test = test_set.test_data.numpy()
> y_test = test_set.test_labels.numpy()
> # Convert to binary labels
> y_train_binary = 2*np.array([y % 2 for y in y_train])-1
> y_test_binary = 2*np.array([y % 2 for y in y_test])-1
>
> #flatten data
> f_X_tr = flatten(X_train) / 255.0
> f_X_te = flatten(X_test) / 255.0
>
> # sample size, and robustness radius
> m, gamma, T = 10000, 1.0, 2
>
> # fit a linear classifier
> print('=============================================================================')
> print('Baseline Linear Classifier:')
> base_clf = LinearSVC(C=10e-7, fit_intercept=False, loss='hinge', tol=1e-5)
> base_clf.fit(f_X_tr, y_train_binary)
> print("0-1 Accuracy - Testing:", 100*base_clf.score(f_X_te, y_test_binary))
>
> # robust accuracy
> y_margin = y_test_binary*(base_clf.decision_function(f_X_te) / np.linalg.norm(base_clf.coef_))
> robust_accuracy = len(y_margin[y_margin >= gamma]) / len(y_margin)
> print('Robust Accuracy - Test:', 100*robust_accuracy)
>
> y_margin = y_train_binary*(base_clf.decision_function(f_X_tr) / np.linalg.norm(base_clf.coef_))
> robust_accuracy = len(y_margin[y_margin >= gamma]) / len(y_margin)
> print('=============================================================================')
>
> # boosting rounds
> print('\nBoosting Robustness')
> clf = []
> for t in range(1,T+1):
>     print("round ", t)
>     # SVM
>     lin_clf = LinearSVC(C=10e-7, fit_intercept=False, loss='hinge', tol=1e-5)
>     lin_clf.fit(f_X_tr, y_train_binary)
>     clf.append(lin_clf)
>
>     if t < T:
>         # Compute margin on fresh sample
>         margin = lin_clf.decision_function(f_X_tr) / np.linalg.norm(lin_clf.coef_)
>         amargin = abs(margin)
>         print("Fraction of Training Data with Robustness at least 2gamma: ", 100*float(len(amargin[amargin >= 2*gamma])/len(amargin)))
>         print(len(amargin[amargin < 2*gamma]), " samples with small margin")
>
>         # Update / Filter training data
>         f_X_tr = f_X_tr[amargin < 2*gamma]
>         y_train_binary = y_train_binary[amargin < 2*gamma]
>
> # Evaluation
> print('\nEvaluation - Test')
> print('number of test samples', len(y_test_binary))
> adv_mistake = 0
> total = len(y_test_binary)
> for t in range(0,T):
>     print('round ', t)
>     amargin = abs(clf[t].decision_function(f_X_te) / np.linalg.norm(clf[t].coef_))
>     y_margin = y_test_binary*(clf[t].decision_function(f_X_te) / np.linalg.norm(clf[t].coef_))
>     adv_mistake += len(y_test_binary[y_margin < -1*gamma])
>     print('adversarial mistakes', len(y_test_binary[y_margin < -1*gamma]))
>
>     f_X_te = f_X_te[(y_margin < 2*gamma) & (y_margin >= -1*gamma)]
>     y_test_binary = y_test_binary[(y_margin < 2*gamma) & (y_margin >= -1*gamma)]
>     print('abstained on data', len(y_test_binary), len(f_X_te))
>
>     # last round
>     if t == T-1 and len(f_X_te)>0:
>         y_margin = y_test_binary*(clf[t].decision_function(f_X_te) / np.linalg.norm(clf[t].coef_))
>         print('classifications in final round', len(y_margin))
>         ee = len(y_test_binary[(y_margin < gamma)])
>         print('extra mistakes', ee)
>         adv_mistake += ee
>
> print('total number of adv. mistakes', adv_mistake)
> print('=============================================================================')
> print('Cascade Robust Accuracy:', 100*float((total - adv_mistake)/total) )
> print('=============================================================================')
> ```

---

### Meta-Review · Area_Chair_BTWy · 2022-08-21

**Recommendation:** Accept
**Confidence:** Certain

**Metareview:**

This paper makes important theoretical, conceptual, and algorithmic contributions to the adversarial robustness literature. I recommend the authors carefully go over reviewer suggestions on expositional clarity. Authors are also encouraged to explicitly discuss the limitations of the proposed approach. In particular, they can provide more discussion on the comparison between U(x) and U^{-1}U(x) and state the need to develop robust verification methods in the paper’s context.

**Award:**

No

---

### Decision · Program_Chairs · 2022-09-14

Accept